# Aerobic microbial life persists in oxic marine sediment as old as 101.5 million years

Yuki Morono [1,2✉], Motoo Ito [1,2], Tatsuhiko Hoshino[1,2], Takeshi Terada[3], Tomoyuki Hori [4], Minoru Ikehara [5], Steven D'Hondt [6✉] & Fumio Inagaki [1,2,7,8✉]

Sparse microbial populations persist from seafloor to basement in the slowly accumulating oxic sediment of the oligotrophic South Pacific Gyre (SPG). The physiological status of these communities, including their substrate metabolism, is previously unconstrained. Here we show that diverse aerobic members of communities in SPG sediments (4.3–101.5 Ma) are capable of readily incorporating carbon and nitrogen substrates and dividing. Most of the 6986 individual cells analyzed with nanometer-scale secondary ion mass spectrometry (NanoSIMS) actively incorporated isotope-labeled substrates. Many cells responded rapidly to incubation conditions, increasing total numbers by 4 orders of magnitude and taking up labeled carbon and nitrogen within 68 days after incubation. The response was generally faster (on average, 3.09 times) for nitrogen incorporation than for carbon incorporation. In contrast, anaerobic microbes were only minimally revived from this oxic sediment. Our results suggest that microbial communities widely distributed in organic-poor abyssal sediment consist mainly of aerobes that retain their metabolic potential under extremely low-energy conditions for up to 101.5 Ma.

[1] Kochi Institute for Core Sample Research, Japan Agency for Marine-Earth Science and Technology (JAMSTEC), 200 Otsu, Monobe, Nankoku, Kochi 783-8502, Japan. [2] Research and Development Center for Submarine Resources, JAMSTEC, 200 Otsu, Monobe, Nankoku, Kochi 783-8502, Japan. [3] Marine Works Japan Ltd, 3-54-1, Oppamahigashi, Yokosuka, Kanagawa 237-0063, Japan. [4] Environmental Management Research Institute, National Institute of Advanced Industrial Science and Technology (AIST), 16-1 Onogawa, Tsukuba, Ibaraki 305-8569, Japan. [5] Center for Advanced Marine Core Research, Kochi University, Nankoku, Kochi 783-8502, Japan. [6] Graduate School of Oceanography, University of Rhode Island, Narragansett Bay Campus, 215 South Ferry Road, Narragansett, RI 02882, USA. [7] Research and Development Center for Ocean Drilling Science, 3173-25 Showa-machi, Kanazawa-ku, Yokohama, Kanagawa 236-0001, Japan. [8] Present address: Mantle Drilling Promotion Office, Institute for Marine-Earth Exploration and Engineering (MarE3), JAMSTEC, Yokohama, Kanagawa, 236-0001, Japan. ✉email: morono@jamstec.go.jp; dhondt@uri.edu; inagaki@jamstec.go.jp

Marine sediment covers ~70% of Earth's surface. It harbors a remarkable microbial population that comprises 12–45% of total microbial biomass or ~0.6–2% of total living biomass on Earth[1–4]. Since the discovery of the subseafloor sedimentary biosphere[5], understanding on the extent, diversity, and physiology of the microbes below the seafloor has greatly expanded[6]. In 2010, Integrated Ocean Drilling Program (IODP) Expedition 329 retrieved sedimentary sequences from the abyssal plain of the South Pacific Gyre (SPG), to examine subseafloor life and habitability in the lowest productivity region of the world ocean[7]. Although their abundance is very low, microbial cells are present throughout the entire sedimentary sequence at all the SPG sites. Their abundance ($2.2 \times 10^2$ to $5.5 \times 10^6$ cells cm$^{-3}$) is one to seven orders of magnitude lower than counts at the same depths in sites of ocean margins and upwelling regions[8]. The presence of dissolved O$_2$, nitrate (NO$_3^-$), phosphate (PO$_4^-$), and dissolved inorganic carbon (DIC) throughout the sedimentary sequence from the seafloor to the volcanic basement indicates that cell abundance and activity are not limited by availability of electron acceptors or dissolved major inorganic nutrients. From the Redfield stoichiometry of net dissolved O$_2$ reduction to net nitrate production in the sediment, the microbial cells have been inferred to consume oxygen coupled to oxidation of marine organic matter at extremely slow rates[8].

Direct evidence of the physiological nature and survival status of microbial cells in this extremely energy-poor setting is previously lacking. A cell must metabolize a certain amount of carbon relative to its own biomass before it can double its size, divide or even sustain a metabolically active state (basal power requirement[9]). Bioenergetic calculations indicate that the microbes in SPG abyssal clay have access to very little power[10]. Very low permeability ($1.1$–$2.0 \times 10^{-17}$ m$^2$ for IODP Site U1365 4H-3 [26.6 meters below seafloor (mbsf)] and $8.9 \times 10^{-18}$ m$^2$ for IODP Site U1370 [37.5 mbsf], respectively)[11], very low estimated pore size of the abyssal clay (~0.02 microns, calculated using above permeability data according to the equation shown in Tanikawa et al.[12]), and thick porcellanite layers above the oldest sampled horizons appear to preclude cell migration into the sampled sediment. Consequently, the sampled communities have likely been trapped in the sediment since shortly after its deposition. The physiological status and growth potential of these buried communities and, more generally, the fractions of these energy-starved subseafloor microbes that are alive, dormant, or dead, have been essentially unknown.

To document the ecophysiology of the microbes that persist in these sedimentary habitats, we set up microaerobic incubation experiments with stable isotope-labeled substrates as tracers for microbial anabolic activities in mini-plugs of ancient (up to 101.5 Ma) pelagic clay (from IODP Sites U1365 and U1370) and relatively young (up to 13 Ma) but extremely oligotrophic calcareous nannofossil ooze (Site U1368, Supplementary Fig. 1). The results—the low relative abundance of spore formers, the rapid initial increases in incubated cell abundance, the high percentages of active cells in the incubations, and the confining nature of their sedimentary habitat—collectively suggest that microbial communities in oxic subseafloor sediment persist in metabolically active form for at least 101.5 million years.

## Results and discussion

**Single-cell anabolic activity with $^{13}$C-, $^{15}$N-labeled substrates.** Single cell-targeted ion imaging analysis of 6986 individual cells by NanoSIMS showed that aerobic microbes from every sample of this oligotrophic oxic sediment actively take up isotope-labeled carbon and nitrogen substrates, even from sediment as old as 101.5 Ma[13] (U1365 9H-3) (Figs. 1 and 2b, c). The number of cellular regions of interests (ROIs) analyzed per incubation sample

ranged from 9 to 210 (Supplementary Data 1). Incorporation of carbon and nitrogen from organic compounds and ammonium was generally seen in the ROIs from all of the microaerobic (~3.3% O$_2$ in the headspace) incubations, with bimodal distribution in the degree of substrate incorporation (Fig. 2b). While initial cell counts were extremely low, ranging from $10^2$ to $3 \times 10^3$ cells cm$^{-3}$ (Fig. 2a)[8], the microbes in the oxic sediment responded rapidly to the substrate injection in microaerobic incubations, increasing total numbers by orders of magnitude and taking up labeled carbon and nitrogen within 68 days of incubation.

In contrast to the microaerobic incubations, incubations of ancient (65.5 Ma[13]) oxic abyssal clay from Site U1370 conducted without adding O$_2$ showed only marginal incorporation of selected substrates with minimal growth of biomass (Supplementary Fig. 2). Although we did not supply additional O$_2$, the oxic sediment contained ~1.5 μM of O$_2$ in its interstitial water[7], which introduced 16.4 nmol of O$_2$ in the incubation vial and reached an equilibrated concentration of not more than 14.6 nM in the sediment. Although the slight increase of biomass at the early stage of the U1370 incubations may have been aerobic microbial growth, there was not enough O$_2$ to sustain aerobic growth for the entire period of incubation. Although the sediment contained abundant dissolved nitrate (~46 μM) and sulfate (~27 μM) in its interstitial water[7], we did not see intensive growth of biomass in later stages of the incubation. This result suggests that metabolic activity hardly revived and did not persist in the absence of added O$_2$ for microbial communities that have been exposed to dissolved O$_2$ for 65.5 million years. This, in turn, suggests that long-time exposure to dissolved O$_2$ over geological time may extinguish anaerobic microorganisms including anaerobic spore-formers (or at least make them hard to revive).

The microbial metabolic response to microaerobic inoculation conditions was heterogeneous; the extent and rate of labeled isotope incorporation varied considerably from one ROI to another (Fig. 2b, c), and from one labeled substrate to another, similarly to the substrate incorporations seen in anaerobic incubations of deep subseafloor coalbed sediment[14]. Of the labeled substrates tested, heterotrophic substrates generally supported biomass growth. However, in the case of nannofossil-bearing clay of U1368 2H-5 sediment, the amino acid-supplemented incubation showed biomass increase only at day 21 of incubation and afterward decreased to the level of original cell abundance, indicating the loss of increased biomass at a later stage of incubation. We observed only marginal autotrophic incorporation of bicarbonate (0–23.6 atom%, in average 0.8 atom%, Fig. 2b), consistent with primary reliance on organic matter as carbon source for biomass production.

**Rates of substrate uptake and biomass generation.** The response to substrate addition, in general, was faster (on average, 3.09 times, Table 1) and more extensive for nitrogen incorporation than for carbon incorporation (Fig. 2c). Ammonium, which was added as a nitrogen source in all incubations except for the incubation with $^{13}$C and $^{15}$N-labeled amino acids, was generally incorporated quickly into cellular biomass. The preferred incorporation of supplemented ammonium over nitrate, which originally existed at ~40 μM in all of the sediment samples[6], was predictable given higher energetic costs of reducing nitrate to the oxidation state of N in amino acids[15]. Although ammonia assimilation requires energy to be incorporated into amino acids[15,16], its incorporation was observed even under the condition supplemented with only ammonium without a carbon source. In addition, the high ammonia incorporation and biomass increase during those incubations suggests at least a minor contribution of chemolithoautotrophic ammonia oxidizers[17].

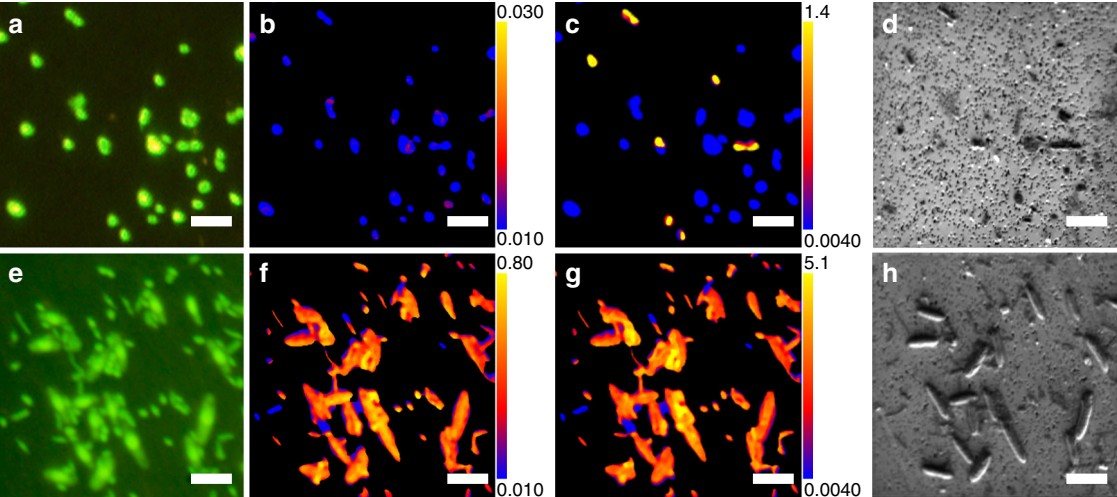

**Fig. 1 $^{13}$C and $^{15}$N incorporation in representative microbial cells.** Cells from incubations of U1365 9H-3 with $^{13}$C-bicarbonate and $^{15}$N ammonium (**a–d**) and $^{13}$C,$^{15}$N-Amino acid mix (**e–h**). (**a**, **e**) SYBR Green I-stained cells under fluorescence microscopy. **b**, **c**, **f**, **g** Ratio images of $^{13}$C/$^{12}$C (**b**, **f**) and $^{12}$C$^{15}$N/$^{12}$C$^{14}$N ratios (**c**, **g**) of the same regions imaged in **a**, **e**, demonstrating locations of $^{13}$C and $^{15}$N incorporation. Color-scale ranges of the ratios are shown as numbers appearing at top and bottom of the color bar. The background membrane region, which is identified by fluorescence images, is excluded from the ratio calculation and shown as black background. **d**, **h**. Secondary electron (NanoSIMS) images of the same regions in **a**, **e**. Bars represent 5 µm. Similar images were processed for obtaining the dataset (Supplementary Data 1) of substrate incorporations for 6986 individual cells.

The growth rates of microbes were calculated based on the rates of biomass growth and substrate incorporation (Fig. 3a, Table 1, and Supplementary Data 2). The biomass-based specific growth rates were generally higher than the substrate-based biomass generation rates. This difference suggests that the active microbial communities utilized carbon and nitrogen compounds in the sediment other than the supplemented isotope-labeled substrates for their biomass growth. Although the total organic carbon content in the sediment is very low (0.01–0.02 weight percent), it exists in bioavailable form but protected under in situ conditions[18]. This amount of total organic carbon is enough to sustain the observed growth of microbes without an additional carbon source; it corresponds to potential microbial biomass[19] of >$10^{10}$ cells cm$^{-3}$, exceeding the original microbial biomass by 7–8 orders of magnitude, and the post-incubation biomass by 4 orders of magnitude.

To document the physiological status of these microorganisms, we calculated the fraction of microbial cells that originally existed in the samples. The calculation was made by using the observed active ROI fractional ratio (number of active ROI versus total ROI analyzed), increase of biomass, and ROI-based sampling depth (the numerical depth of sampling that the ROI-based analysis covered). The observed "active" ROIs ranged from 18.4 to 100% of the total ROIs analyzed, giving an average of 92%. The minimum original active fractions average 77.7%, ranging from 24.1 to 99.1% (Table 1 and Supplementary Data 2). Even in the sample of oldest sediment (U1365 9H-3, 101.5 Ma[13]), the revivable heterotrophic population was 39.6–99.1% of the original community. These microaerobic incubations differ notably from the previously published anaerobic incubation[14] by rapidly and consistently increasing in biomass after the addition of substrates. The biomass-based doubling time ($^{b}T_{d}$) of microbes in these micro-aerobic incubations (Supplementary Data 2) averaged 4.9 days, ranging from 1.4 to 17.8 days. Incubations with the oldest sediment (U1365C 9H-3) showed significantly longer $^{b}T_{d}$ (in average 9.8 days, pairwise $t$-tests, $p < 0.05$) than the incubations with younger sediment. The incubations with U1365C 8H-2 (95.4 Ma[13]) showed slightly longer but statistically insignificant $^{b}T_{d}$ values than the U1368 sediment samples (Table 1).

"Inactive" ROIs, which have no detectable incorporation of supplemented substrates, represent either (i) dormant cells or (ii) dead cells (necromass). Although recycling of biomolecules from necromass is a well-known anabolic strategy in marine sediment[20,21], necromass provides only a small fraction of the maintenance power demand for the microbial community in subseafloor sediment[22]. In our amino-acid-amended incubations, an apparent biomass decrease after an initial increase in the sample of U1368D 2H-5 (Fig. 2a) demonstrates the generation of new biomass and the remineralization after cell death (necromass) at a later incubation stage. The absence of substrate incorporation in the remaining "inactive" cells (accounting for 73.3% of total cells) at the 557-day timepoint suggested that they should have been inactive throughout the incubation period. Therefore, we infer that these cells comprise living (or at least degradation-resistant) but dormant biomass not revived by the addition of amino acids. This pattern of biomass change was not obvious or absent in the incubations with other substrates.

**Comparison of 16S rRNA gene profiles to substrates**. We investigated the 16S rRNA gene-based taxonomic composition of microbial cells sorted by fluorescence-activated cell sorting (FACS) from each incubation. Although pilot studies showed the community composition in anoxic sediment outside the SPG[23], near-seafloor SPG sediment (<10 cmbsf)[24], and SPG surface seawater[25], diversity of microbial community in SPG sediment more than a meter below seafloor has not been reported. This absence of community composition information is partially due to the very low cell abundance and the nature of the sediment samples (dominated by fine-grained clay and silt with negligible sand[26]) that may have adsorbed released DNA from lysed cells during the DNA extraction process. For this study, we separated and sorted cells from the sediment to obtain community taxonomic composition under all of the incubation conditions, including initial sediment samples with very low biomass (~$10^{2}$ cells cm$^{-3}$) (full dataset can be found in Supplementary Data 2 and detailed methods to eliminate procedural contaminations are found in "Methods" section). The communities of all samples were dominated by bacterial sequences. Dominant bacterial groups included

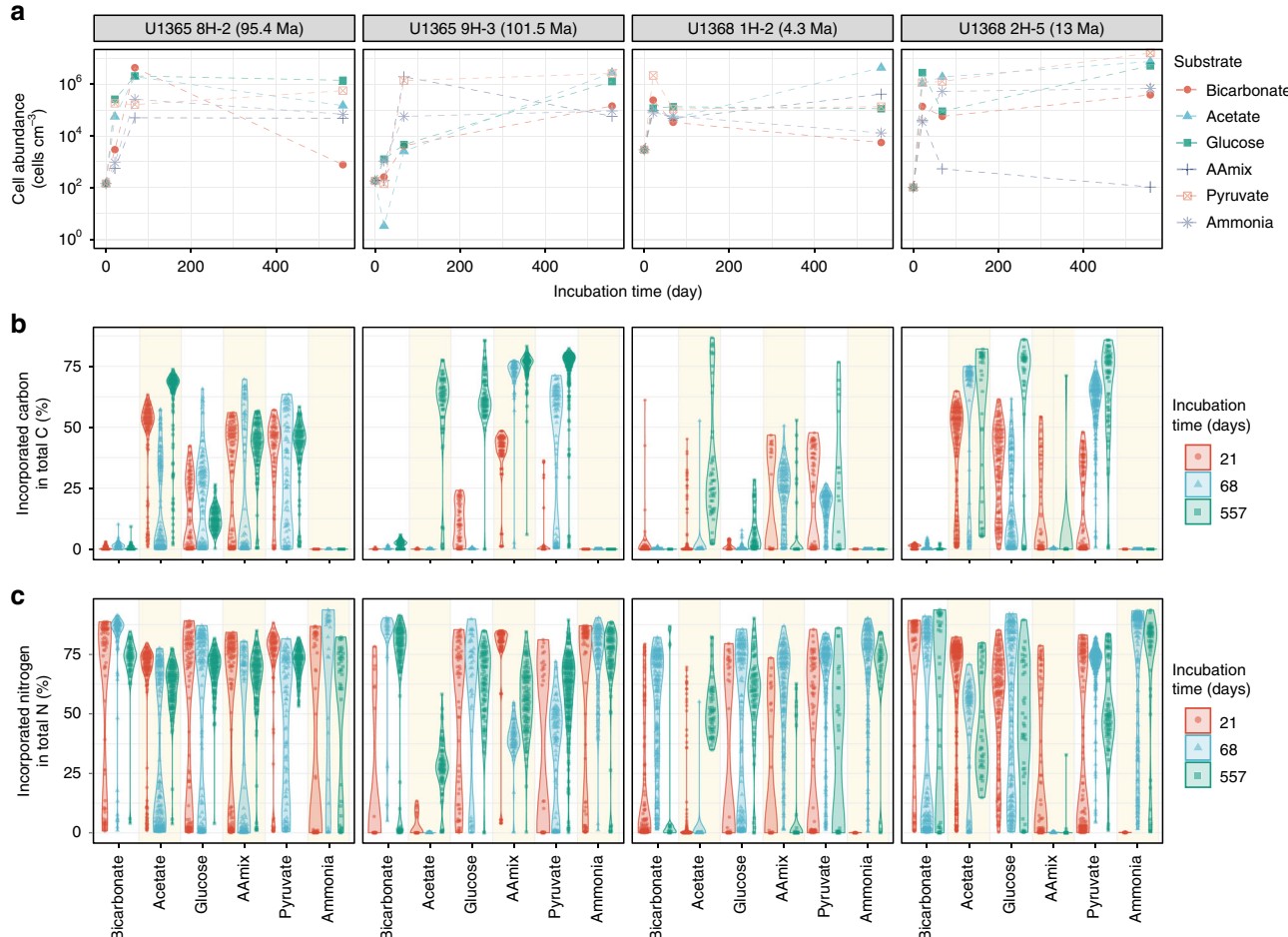

**Fig. 2 Microbial responses to addition of carbon and nitrogen substrates.** Plots for samples used for the incubations are aligned vertically. **a** Plots of cell abundance during incubation. Cell abundances for Incubation time 0 are abundances in the sediment samples before incubation was initiated. For the samples incubated with carbon substrates (bicarbonate, acetate, glucose, and pyruvate), ammonia was added as nitrogen source. The incubation labeled "Ammonia" received ammonia as the nitrogen source with no additional carbon. $n = 76$ samples (one for Incubation time 0 and three for Incubation sampling points [days 21, 68, and 557] per substrate for each sediment sample). **b**, **c**. Incorporation of carbon- (**b**) and nitrogen- (**c**) substrates by microbes identified by NanoSIMS cellular ROIs (Supplementary Data 1). Substrate incorporation for each ROI was plotted over kernel density violin plots.

*Actinobacteria, Bacteroidetes, Firmicutes, Alphaproteobacteria, Betaproteobacteria, Gammaproteobacteria,* and *Deltaproteobacteria* (Fig. 3b, c) with a minor fraction of *Chloroflexi* (0– 2.6%). A small fraction (~7%) of Terrestrial Hot Spring Group of thermophilic *Archaea*, which was recovered from sediment overlying relatively young (13.5-Ma) basement on the East Pacific Rise flank, was detected in sample U1368D 2H-5 (Supplementary Data 2). The absence of obligately autotrophic taxa and ammonia-oxidizing microorganisms confirmed the marginal incorporation of $^{13}C$-labeled bicarbonate and the dominance of heterotrophic ammonia anabolism in the incubations. Also, it should be noted that we found a large fraction (54.9%) of *Cyanobacteria* in the incubation of U1368 2H-5 supplemented with ammonia at the time point of 557 days. Most of these *Cyanobacteria* (99.4%) were *Chroococcidiopsis,* hypolithic cyanobacteria with high tolerance to extreme environmental condition[27,28], and minor fraction of non-photosynthetic unclassified *Obscuribacterales*. The abundant distribution of *Chroococcidiopsis* in various extreme environments on Earth, including marine environments, are consistent with their survival in this sediment sample. However, the detailed mechanism for *Chroococcidiopsis* growth under dark incubation condition is unclear.

In general, microbial community composition in the incubations did not closely relate to the specific substrate added (Fig. 3c).

This is consistent with the suggestion from NanoSIMS-based results that the active microbial community utilizes native carbon and nitrogen compounds other than the substrates added to the incubation, and additionally suggest that the added substrates just stimulated the activity of microbes for their utilization of native carbon and nitrogen compounds. Rather the community composition showed dependence on incubation time, which suggests that different taxa responded to incubation at different timescales (Fig. 3c).

**Abundance of spore formers.** The mechanisms that allow microbial survival at extraordinarily low energy fluxes in sub-seafloor sediment are largely unexplored. Spore formation has been suggested as one of the possible mechanisms for such long-term survival of microorganisms[29–31]. Spore-forming lineages, however, were only a minor constituent of the microbial communities. *Firmicutes* related to thermophilic spore-forming microorganisms (*Thermincola*[32] and *Carboxydocella*[33]) were detected at 0–12% (averaging 0.87 and 0.92%, respectively) throughout the samples and incubations. A small fraction (~2.5%) of the populations in incubations of the oldest sediment sample (U1365C 9H-3) is related to the genus *Paenibacillus*[34], mesophilic spore-forming bacteria. This low relative abundance

**Table 1 Growth characteristics of incubations.**

| Sample | Substrate | Incubation time (days) | Biomass-based maximum specific growth rate ($\mu_B$, day$^{-1}$) | Doubling time ($^bT_d$, day) | Carbon substrate incorporation-based biomass generation rate ($^C\mu_S$, day$^{-1}$) | Nitrogen substrate incorporation-based biomass generation rate ($^N\mu_S$, day$^{-1}$) | Estimated original active fraction |
|---|---|---|---|---|---|---|---|
| U1365 8H-2 | Acetate | 21 | 0.284 | 2.4 | 0.032 | 0.052 | nm |
|  | AAmix | 21 | 0.086 | 8.0 | 0.022 | 0.047 | 96.6% |
|  | Bicarbonate | 21 | 0.152 | 4.6 | 0.000 | 0.060 | 82.1% |
|  | Glucose | 21 | 0.358 | 1.9 | 0.010 | 0.047 | nm |
|  | Ammonia | 21 | 0.111 | 6.3 | nm | 0.042 | 47.4% |
|  | Pyruvate | 21 | 0.341 | 2.0 | 0.025 | 0.063 | nm |
| U1365 9H-3 | Acetate | 21 | 0.039 | 17.9 | 0.002 | 0.003 | 99.1% |
|  | AAmix | 21 | 0.136 | 5.1 | 0.023 | 0.073 | 98.9% |
|  | Bicarbonate | 21 | 0.046 | 15.1 | 0.000 | 0.027 | 38.0% |
|  | Glucose | 21 | 0.092 | 7.5 | 0.006 | 0.043 | 89.0% |
|  | Ammonia | 21 | 0.084 | 8.2 | nm | 0.059 | 92.4% |
|  | Pyruvate | 21 | 0.131 | 5.3 | 0.010 | 0.037 | 87.2% |
| U1368 1H-2 | Acetate | 21 | 0.175 | 4.0 | 0.004 | 0.006 | nm |
|  | AAmix | 21 | 0.166 | 4.2 | 0.021 | 0.032 | 88.7% |
|  | Bicarbonate | 21 | 0.211 | 3.3 | 0.001 | 0.014 | 93.9% |
|  | Glucose | 21 | 0.176 | 3.9 | 0.001 | 0.037 | 67.1% |
|  | Ammonia | 21 | 0.157 | 4.4 | nm | 0.020 | 86.8% |
|  | Pyruvate | 21 | 0.315 | 2.2 | 0.023 | 0.053 | 74.0% |
| U1368 2H-5 | Acetate | 21 | 0.441 | 1.6 | 0.027 | 0.050 | nm |
|  | AAmix | 21 | 0.286 | 2.4 | 0.013 | 0.022 | 24.1% |
|  | Bicarbonate | 21 | 0.344 | 2.0 | 0.000 | 0.068 | nm |
|  | Glucose | 21 | 0.487 | 1.4 | 0.022 | 0.042 | nm |
|  | Ammonia | 21 | 0.278 | 2.5 | nm | 0.024 | nm |
|  | Pyruvate | 21 | 0.445 | 1.6 | 0.014 | 0.031 | nm |

*nm* not measurable.

of known spore formers suggests that the revival of spores (germination) was very limited in the incubations.

## Methods

**Materials**. Sediment samples used in this study were collected by IODP Expedition 329 in the South Pacific Gyre (SPG; Supplementary Fig. 1)[7]. The samples collected from Site U1365 (23° 51.0377′S 165° 38.6502′W) are zeolitic and/or metalliferous pelagic clays, those from Site U1368 (27° 54.9920′S, 123° 09.6561′W) are calcareous nannofossil oozes, and those from Site U1370 (41° 51.1267′S, 153° 06.3674′W) are black metalliferous clay, containing light yellowish brown clay-bearing nannofossil ooze. The water depths of these sites are respectively 5697, 3739, and 5075 meters below sea level (mbsl) at U1365, U1368, and U1370. Whole-round core samples of U1365C 8H-2 (obtained from 68.9 meters below seafloor [mbsf], 95.4 Ma), U1365C 9H-3 (74.5 mbsf), U1368D 1H-2 (1.6 mbsf), 1368D 2H-5 (14.7 mbsf), and U1370F 7H-6 (62.9 mbsf) were used for incubation experiments as described in the following section. Those samples were from horizons with no observable coring disturbance by visual core description[35]. Drilling fluid contamination assessment by chemical tracer revealed minimal drilling contamination of the samples (≤1 cell/ g-sediment for U1365C 8H-2 and 1368D 2H-5, below detection to 0 cell/g-sediment for U1365C 9H-3, U1368D 1H-2, and U1370F 7H-6)[35]. All Expedition 329 data are archived and available online in the IODP database (http://iodp.tamu.edu/ tasapps) and archived online in the IODP Expedition 329 Proceedings[7]. Additional sediment age estimation is available in reports by Dunlea et al.[13] and Alvarez Zarikian et al.[36].

**Incubation experiments**. To identify autotrophic and heterotrophic microbial populations, as well as their potential rates for growth and substrate uptake, sediment samples were incubated with stable isotope-labeled substrates ($^{13}C_6$-glucose, $^{13}C_2$-acetate, $^{13}C_3$-pyruvate, $^{13}C$-bicarbonate, $^{13}C$-$^{15}N$-amino acids mix [mixture of 20 Amino Acids], and $^{15}N$-ammonium). The incubation experiments were initiated onboard during expedition. Avoiding outer edges of the sediment cores (which are likelier to be contaminated by drill fluid), each 15 cm$^3$ sample was taken from an interior portion of a core with a sterile tip-cut 30 mL syringe, and placed in a 50 cm$^3$ sterile glass vial (Nichidenrika-Glass Co. Ltd.) sealed with a sterile rubber stopper and a screw cap, followed by flushing with 0.22 μm filter-sterilized nitrogen gas and storage at 10 °C. All of the sterilization was done by autoclaving of the materials at 121 °C for 20 min. Given O₂ concentrations in interstitial water during Expedition 329 (~70 μM for U1365C 8H-2 and 9H-3, ~150 μM for U1368D 1H-2, ~130 μM for 1368D 2H-5, and ~1.5 μM for U1370F 7H-6)[7], we set O₂ concentration in the headspace of vials (other than U1370F 7H-6) at ~3.3% (v/v), which corresponds to

the aqueous concentration of oxygen at ~43 μM (assuming salinity of sea water), by adding 0.22 μm filter-sterilized air. The labeled substrates were injected (15 μM of $^{13}C$-labeled substrates, and 1.5 μM of $^{15}N$-labeled ammonium, dissolved in 50–100 μL of sterile water) onto each subcore sample by syringe and needle and incubated at 10 °C (Supplementary Fig. 3a). All reagents and gas components, including air used for sample preparation, were filtered through 0.22 μm syringe top filter. After setting up the incubations, one of the vials from each set of incubation had no substrate added, and a sediment split from the same sample was fixed by adding equal volume (15 mL) of 4% paraformaldehyde (PFA) in PBS solution for 5 h at 4 °C (time point T0). At each of three time points (T1: ~3 weeks [21 days], T2: 6 weeks [68 days], T3: 18 months [557 days] after starting incubation), vials were opened and sediment samples were fixed with equal volume (15 mL) of 4% PFA in PBS solution for 5 h at 4 °C. Fixed samples were frozen at −80 °C. After storage, they were washed twice with PBS and preserved in PBS/ethanol (1:1 [v/v]) at −20 °C until analysis.

**Cell enumeration and selective sorting onto the membrane**. To efficiently analyze substrate incorporation into microbial cells with NanoSIMS, cells were separated from their sediment matrix and fluorescence-activated cell sorting (FACS) was conducted to concentrate and purify cells in a small area for analysis (~0.5 mm², Supplementary Fig. 3b)[14,37]. Cell separation, enumeration, and FACS were all conducted in the clean-booth and clean-room facilities at the Kochi Institute for Core Sample Research, Japan Agency for Marine-Earth Science and Technology (JAMSTEC).

Cell separation, microscopy, and sorting procedures followed the method of Morono et al.[37] with modifications. Eight milliliters of fixed slurry (1/3 [v/v] sediment in ethanol-PBS solution) was mixed with the same volume of 2.5% NaCl solution, followed by centrifugation at 4500 × g for 15 min, after which the supernatant was discarded and the pellet resuspended by adding 2.5% NaCl solution to be 20 mL of sediment slurry. The sediment slurry was added with 2.5 mL each of detergent mix[38] (100 mM EDTA, 100 mM sodium pyrophosphate, 1% [v/v] Tween 80) and methanol, then vigorously shaken for 60 min at 500 rpm using a Shake Master (Bio Medical Science, Tokyo, Japan). After shaking, the sediment slurry was sonicated (Bioruptor UCD-250; COSMO BIO) in an ice bath for 20 cycles of 30 s at 200 W on and 30 s off. The processed slurry was then carefully layered onto a manually layered high-density cushion solution consist of (from top) 4 mL of 30% [v/v] Nycodenz, 4 mL of 50% [v/v] nycodentz, 4 mL of 80% [v/v] Nycodentz, and 4 mL of 67% [w/v] of sodium polytungstate. Samples were centrifuged at 4000 × g for 120 min, after which the supernatant, including the high-density layer(s), was carefully removed and transferred to a separate vial. Cells in the supernatant were trapped onto an Anopore Inorganic Membrane (Anodisc,

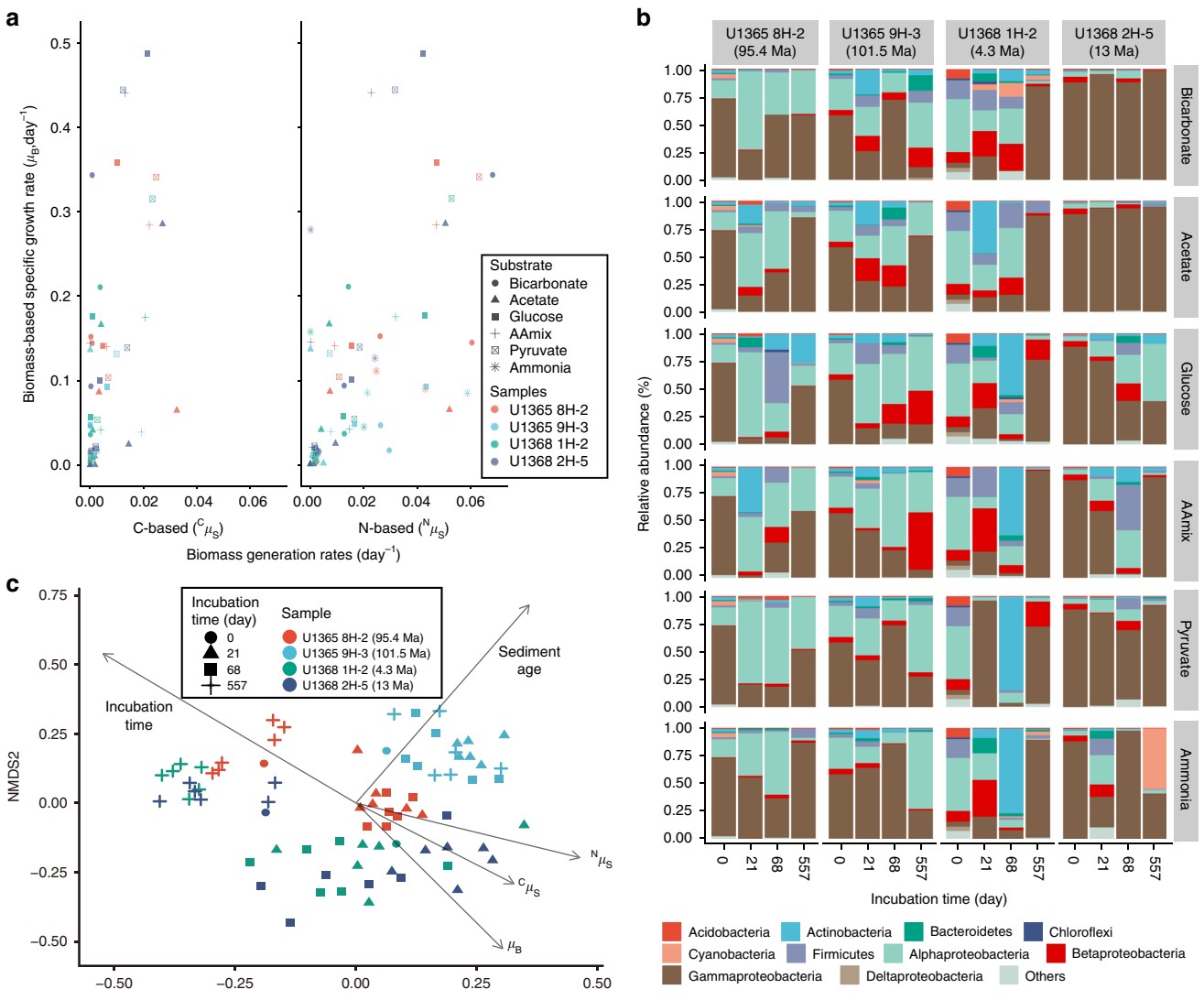

**Fig. 3 Microbial growth response to the addition of substrates. a** Scatter plots of microbial growth characteristics calculated from biomass (biomass-based specific growth rate, $\mu_B$ [day$^{-1}$]), carbon and nitrogen substrate (substrate-based biomass generation rates, $^C\mu_S$ [day$^{-1}$] for carbon and $^N\mu_S$ [day$^{-1}$] for nitrogen, respectively. $n = 76$ samples were analyzed for the calculation of $\mu_B$, $n = 4298$ and $n = 6028$ cells were analyzed for the calculation of $^C\mu_S$ and $^N\mu_S$, respectively [of totally 6253 cells analyzed, non-zero values of $^C\mu_S$ and $^N\mu_S$ were used for this plot]). **b** Relationship of community composition to addition of substrates along the time of incubation. The community compositions at "Incubation time 0" show the compositions before incubation. **c** Non-metric multidimensional scaling (NMDS) ordination plot based on Bray–Curtis dissimilarities of community compositions at different incubation conditions and time points (stress = 0.209). Arrows represent environmental variables that were significantly ($p < 0.005$, permutations = 10000) related to the ordination. $n = 76$ samples (one for Incubation time 0 and three for incubation sampling points [days 21, 68, and 557] per substrate for each sediment sample).

Whatmann, Kent, UK), washed with TE buffer, and then stained with 100 μL of SYBR Green I staining solution. After staining for 5 min, the SYBR-stained cells were washed with 2 mL of TE buffer, and then the membrane was placed into a 50 mL centrifuge tube containing 5 mL of TE buffer. Cells were detached from the membrane by sonication at 20 W for 10–30 s using a Model UH-50 Ultrasonic Homogenizer (SMT Co. Ltd., Tokyo, Japan) and concentrated to be 1.5 mL by centrifugation at $7000 \times g$ for 10 min and discarding 3.5 mL of the supernatant. Part (0.5 mL) of the stained cell suspension was filtered onto 0.22-μm pore size black polycarbonate membrane (Isopore GTBP02500; Millipore) and used for counting microbial cells by the fluorescence color-based discriminative cell enumeration method[39,40]. Cells were sorted following the flow cytometry protocol of Morono et al.[37] directly from the sorter onto NanoSIMS-compatible 0.2-μm polycarbonate filters coated with indium tin oxide (ITO)[14,41] and non-coated membrane (Isopore GTBP02500; Millipore). ITO coating on polycarbonate membranes (Isopore GTBP02500; Millipore) was prepared by a sputtering deposition technique at Astellatech Co. Ltd. The sorted cells on the non-coated membrane were stored at −20 °C until DNA extraction.

**NanoSIMS analysis of single cell-image acquisition and data processing.** Cell targets were identified by fluorescence of SYBR Green I stain and marked on

NanoSIMS membranes with a laser dissection microscope (LMD6000; Leica Microsystems) for ease of rediscovery on the NanoSIMS (an example is shown in Supplementary Fig. 3b). Microbial cells that incorporated stable isotope-labeled substrates were analyzed using NanoSIMS 50L (AMETEK Co. Ltd., CAMECA BU) at the Kochi Institute for Core Sample Research, JAMSTEC. Samples on the ITO-coated polycarbonate membrane were pre-sputtered at high beam currents (30 pA/s/μm$^2$) before measurement. The $^{12}C^-$, $^{13}C^-$, $^{12}C^{14}N^-$, $^{12}C^{15}N^-$ and $^{32}S^-$ secondary ions were collected and measured in parallel at a mass resolution of 8000 that is sufficient to separate $^{13}C^-$ from the $^{12}CH^-$ and $^{12}C^{15}N^-$ from $^{13}C^{14}N^-$. Samples were measured using a 1–2 pA Cs$^+$ primary beam that was rastered over $25 \times 25$ μm field of a $256 \times 256$ pixels with a counting time of 5 ms per pixel. Recorded images and data were processed using CAMECA WinImage software and OpenMIMS plugin[42] in ImageJ[43] distribution of Fiji[44]. Different scans of each image were aligned to correct image drift during acquisition. Final images were created by adding the secondary ion counts of each recorded secondary ion from each pixel over all scans. Intracellular carbon and nitrogen uptake from stable isotope-labeled substrates was calculated by drawing regions of interest (ROI) on CN$^-$ images (recognizing cells in the images) and calculating $^{13}C/^{12}C$ and the $^{15}N/^{14}N$ ratio (calculated from the $^{12}C^{15}N/^{12}C^{14}N$ ratio). Instrumental mass fractionation of NanoSIMS analysis was calibrated by the conversion factor

obtained by comparing carbon and nitrogen isotopic ratios of *E. coli* cells of varying isotopic enrichments measured at single cells with NanoSIMS and at bulk with an elemental analyzer/isotope ratio mass spectrometer (EA/IRMS, FlashEA 1112/DeltaPlus Advantage, Thermo Fisher Scientific). Concentration of bicarbonate (DIC) in the original sample determined on board[7] was used to calculate substrate incorporation ratio (atom %) for bicarbonate in single cells.

**Biomass and isotope calculations**. Isotope incorporation data analysis and display as violin plots of the kernel density function were done using R[45] with the "ggplot2"[46], "cowplot"[47], "ggsci"[48], "scales"[49] packages. The "active" cell ROIs, those incorporated $^{13}$C- and/or $^{15}$N-labeled substrates, were determined by their isotopic ratio exceeding the 99.7% confidence interval of background carbon and nitrogen isotopic abundance of polycarbonate membranes (1.24 atom % for $^{13}$C and 0.446 atom % for $^{15}$N). As measures for the rate of microbial biomass synthesis, we calculated the biomass-based specific growth rate and the substrate incorporation-based biomass generation rates. The biomass-based specific growth rates ($\mu_B$, Eq. (1)) were calculated from the abundance of cells in the sediment samples at the start of incubation $X_0$ and the abundance at the incubation period $t$ ($Xt$). The substrate incorporation-based biomass generation rates[14,50] ($^C\mu_S$ and $^N\mu_S$, Eqs. (2) and (3), for carbon and nitrogen substrates, respectively) were calculated from the fractional abundance of isotope label in cellular biomass, where $\mu_S$ is the biomass generation rate (encompassing both cell maintenance and generation of new cells), $t$ is the length of the incubation, $F_{label}$ is the labeling strength, $F_t$ is the single-cell NanoSIMS measurement, and $F_{nat}$ is the natural abundance. For the calculation of $^C\mu_S$ and $^N\mu_S$, a conservative approach was used by only including ROIs where either $^{13}$C or $^{15}$N ratio was above the 99.7% confidence interval of backgrounds shown above. These rate calculations assumed that the carbon and nitrogen for the biomass generation were all derived from the substrates.

$$\mu_B = \frac{\ln X_t - \ln X_0}{t} \quad (1)$$

$$^C\mu_S = \left(-\ln\left(1 - \frac{(^C F_t - {}^C F_{nat})}{(^C F_{label} - {}^C F_{nat})}\right)\right)/t \quad (2)$$

$$^N\mu_S = \left(-\ln\left(1 - \frac{(^N F_t - {}^N F_{nat})}{(^N F_{label} - {}^N F_{nat})}\right)\right)/t \quad (3)$$

To document the physiological status of microorganisms in the original sediment samples, the fraction of microbes that originally existed in the sediment samples ($f_0$) was calculated from the observed active ROI fractional ratio ($f_t$), a factor of biomass increase ($A$) by following equation (Eqs. (4)–(6)).

$$X_t = A X_0 \quad (4)$$

$$X_0(1 - f_0) = X_t(1 - f_t) \quad [f_t < 1, X_t(1 - f_t) < X_0] \quad (5)$$

$$f_0 = 1 - A(1 - f_t) \quad (6)$$

If the number of ROIs was not enough to give enough sampling depth for fulfilling the Eq. (5), the $f_0$ could not be calculated and was shown as N.A.

**DNA extraction and sequencing**. Microbial cells on non-coated polycarbonate membrane were lysed using a protocol modified after Morono et al.[51] at ISO Class 1 clean air environment in super-clean room of JAMSTEC-KOCHI[52]. Sorted cells on the membrane were lysed by three-step alkaline treatments to minimize fragmentation of extracted DNA. First, five microliters of denaturation buffer (200 mM KOH, 20 μM EDTA) was added to the tiny piece of membrane with sorted cells, incubated at room temperature for 5 min, followed by adding 5 μL of neutralization buffer (200 mM HCl, 300 mM Tris-HCl [pH7.5]). The liquid part was recovered for the second step, which was adding another 5 microliters of denaturation buffer, heating at 70 °C for 5 min, and adding 5 μL of neutralization buffer, followed by recovery of the liquid part and pooling with the first extract. The third step was adding an additional 5 μL of denaturation buffer, heating at 70 °C for 10 min, and adding 5 μL of neutralization buffer, followed by recovery of the liquid part. Heating was done with an Applied Biosystems Veriti Thermal Cycler (Thermo Fisher Scientific). The lysis of cells was confirmed by checking processed membrane with fluorescence microscope after staining with SYBR Green I (Thermo Fisher Scientific). Negative control extractions (8 separate extractions) were performed by using polycarbonate membrane without sorted cells. The resultant DNA extract was then used directly for first-round PCR amplification with universal primers targeting the V4 region of the 16S rRNA gene (U515F: 5′-TGY CAG CMG CCG CGG TAA-3′, U806R: 5′-GGA CTA CHV GGG TWT CTA AT-3′). Also, PCR No-Template-Control reactions (PCR-NTCs) run without template DNA (6 separate reactions) were performed. First-round PCR amplicons were quantified and purified via gel electrophoresis and then indexed and barcoded in the second PCR. The DNA sequencing was performed on a MiSeq Illumina platform following the methods outlined by Hoshino and Inagaki[53]. The obtained sequence reads were quality trimmed and filtered by FASTQ Toolkit in Basespace (Illumina). Resultant paired-end reads were processed using mothur[54] following the Mothur software package (v.1.35.0) of Illumina MiSeq Standard Operating Procedure[55]. All

sequence reads from negative control extractions and PCR-NTCs, along with sequences from all samples were used to generate operational taxonomic units (OTUs) at 97% similarity using usearch 10.0 program[56]. To remove sequences that may be of exogenous origin[57], the most conservative way to remove contaminant sequences considers all taxa that are identified in the laboratory controls as contaminants. This approach has the advantage of eliminating any false-positive discoveries of novel taxa; however, it has the disadvantage of introducing false-negative taxa, which may be of interest. For example, microbial species that are ubiquitous in environments may be excluded from a sample. To circumvent this issue, we employed a "probabilistic" approach to examining the microbial composition based on the consistency (i.e., variance) of the relative distribution of the taxonomic assignments or OTUs across the entire set of samples[41]. This approach compares the proportion of OTUs present between negative controls (i.e., extracted negative controls or PCR-NTCs) and samples, and determines the significantly more abundant OTUs in negative controls (ANOVA analysis) to be identified as contaminants and excluded from the sequence assemblage. By accounting for the consistency (variance) at which any taxon or OTU is found in the negative controls or set of experimental samples, a more nuanced view of the overall data set can be achieved. For the ANOVA analysis, we used Qiime[58] script group_significance.py to estimate a likelihood based on observed abundances for conditions (negative controls and experimental samples). This process was conducted sequentially on extraction negative controls and PCR-NTCs, and the OTUs that have significantly higher ($p < 0.05$, ANOVA) abundance in the extraction negative controls and the NTCs than in the incubation samples were identified and removed from further analysis. This read set was taxonomically classified by using the mothur utility package with the SSURef_NR99_123_SILVA database[59,60]. The total sequences and relative abundance of taxa in the final sequence dataset are provided in Supplementary Data Set 2. The relative abundance of bacterial and archaeal OTUs at the genus level were exposed to non-metric multidimensional scaling with the metaMDS function of R software package vegan[61]. A Bray–Curtis distance matrix was used as dissimilarity index.

**Reporting summary**. Further information on research design is available in the Nature Research Reporting Summary linked to this article.

## Data availability

DNA sequence data are deposited in the DNA Data Bank of Japan (DDBJ) under the accession code of DRA007733. The authors declare that the data supporting the findings of this study are available within the paper and the Supplementary Information.

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

## Acknowledgements

The samples used in this study were collected during IODP Expedition 329 by the *JOIDES Resolution*. We deeply appreciate the work of the shipboard crews, operational team members, and shipboard scientists of IODP Expedition 329. The authors thank Satoko Tanaka, Sae Fukunaga, Megumi Becchaku, and Tomoka Fukumi for their technical assistance for this study. This study was partly supported by the Japan Society for the Promotion of Science (JSPS) Grant-in-Aid for Scientific Research (no. 24687004, 15H05608 and 19H00730 to Y.M., no. 15K14907 and 17H03956 to T.H., and no. 26251041 and 19H05503 to F.I.), JSPS Strategic Fund for Strengthening Leading-Edge Research and Development (to JAMSTEC and F.I.), the JSPS Funding Program for Next Generation World-Leading Researchers (no. GR102 to F.I.), and the US National Science Foundation (through the Center for Dark Energy Biosphere Investigations (C-DEBI) [grant NSF-OCE-0939564 to S.D.]). This is a contribution of the Deep Carbon Observatory (DCO) and the Earth 4D: Subsurface Science and Exploration, CIFAR. It is C-DEBI publication 532.

## Author contributions

Y.M., F.I., and S.D. designed the study; S.D. and F.I. co-led IODP Expedition 329; Y.M. and F.I. carried out incubation of sediment samples with stable isotope substrates; Y.M. and T.T. prepared the samples for NanoSIMS analysis; M.I. and Y.M. conducted the NanoSIMS analyses on single cells; Y.M. and T.T. extracted the DNA; Y.M., Ta.H., and To.H. sequenced the DNA and analyzed the sequencing data; Y.M. S.D. and F.I. were the primary authors of the manuscript with input from all other co-authors.

## Competing interests

The authors declare no competing interests.
