## [Peer Review File · Nature Communications]

Peer Review File - Reviewers' comments first round:

Reviewer #1 (Remarks to the Author):

Morono et al present a manuscript where they used cutting-edge stable isotope labeling techniques and single cell visualization of substrate utilization, from enrichment cultures deriving from deep subseafloor oxygenated sediments that have ages ranging from 12 - 101.5 million years old. The conclusion is that the authors were able to revive bacteria from such ancient samples, which would indeed be truly stunning. However, in my opinion there is cause for concern regarding contamination, before these conclusions can be believed beyond a reasonable doubt.

Unfortunately, there is no in situ data presented that is demonstrating presence of the enriched bacteria in the actual samples. This is needed to confirm that what grew was actually in the samples. Many of the bacteria enriched are related to typical contaminants. For example, Alphaproteobacteria, Betaproteobacteria and Firmicutes are possible contaminants (e.g., Bacillus spores from soil, Burkholderiales, and Rhizobium, they are ubiquitous in labs). Furthermore, there are a lot of cyanobacteria showing up in some of the samples - this must be contamination. How else you can explain that? In some cases the cyanobacteria are almost 50% of the sequences (e.g., 557 day timepoint). Some Chloroflexi are possibly authentic, since studies focusing on in situ communities of samples from the SPG and similar sites from the Atlantic showed that there should be a lot of Chloroflexi in these settings. Since the cell counts are very low for the authors sediments (10^2 - 10^3 cells gram) the possibility for contamination is extremely high. I know that the authors are using the best and most state of the art facilities and controls for contamination, but still the fact that what was enriched differs so much from the earlier SPG studies on in situ communities makes me wonder what happened over these long incubation times. Reading the methods, I have identified one possible source of contamination: the rubber stoppers and screw caps. It seems that they were not sterilized (at least this is not written). This alone could have introduced contaminants into all the incubations. Contamination from the outside of the rubber stopper can also be introduced when puncturing with a syringe (to add the labeled substrates). It's not clear how this was dealt with.

In their defense, it seems that the authors already looked at the OTUs they found in the enrichments and compared them against OTUs that they have sequenced from extraction blanks and/or other sources of contamination in the lab. This is a good idea. That should also help to show how much of what grew was contamination and how much was from the actual samples. I presume that most of the contaminants are Betaproteobacteria, Alphaproteobacteria, Firmicutes, and Cyanobacteria. If that is the case, how confident are you that OTUs from those same groups in your incubations are not contaminants? Are they at least from different genera? If they are all from the same genus, but different OTUs that is probably a contaminant in my opinion. If they are from different genera it is more convincing. There should be a long discussion on this in the manuscript.

Given these concerns, in my opinion there thus remains a critical aspect to be checked: do the substrate utilization rates observed by the authors derive from in situ microbes or those derived from bacteria introduced by contamination during experimental setup (e.g., from the non-sterilized rubber stoppers). The low rates of activity seen by the authors do not themselves serve as a suitable check for this, because if you take a low number of contaminating bacteria and put them under nutrient depleted conditions then they should also grow slowly (or not at all).

Don't get me wrong, I think it's entirely possible that some of the bacteria the authors observe could be from the actual samples (e.g., Chloroflexi) and actually be waking up after a long sleep. But, currently it is impossible to tell what is from the sample and what is a contaminant. Can the authors just do a 16S survey from DNA from the frozen samples to confirm that the OTUs that are enriched are also present in situ? That would basically provide the information needed, and satisfy all of the concerns I have outlined above.

Reviewer #2 (Remarks to the Author):

This paper describes the incubations of deep oxic sediments and the observation that cells in these sediments readily take up substrates, suggesting that life is preserved in oxic sediments for long time periods. The experiments are well done and explained. This work adds to important understanding of the deep biosphere, but moreover raises more questions than it answers because few of the lineages recovered have any known long term resting form. Because of this, I think it is the proper level for a Nature Communications piece, since it should be widely read and cited for future work.

Major comments:

The current discussion of controls is insufficient in the main text and should be improved.

For the cells that did not respond to uptake, but appeared to remain dormant, what was their isotopic value? Do you have any idea on the ionic content? Recent work shows that LIBS is able to show that ions are lost when membranes are damaged - is there anything more that can be done to explain how/why these cells remain dormant?

During uptake was any change seen in average cell size or shape?

I know there is a word limit, but noting the cyanobacteria observed in Ammonia incubations may be worthy.

I in general dislike the cell abundance graphs used. There are only a few datapoints and the trend between datapoints is truly unknown. If the authors can think of presenting them in any other way it would be useful.

Minor comments:

line 37 "catch up" is too informal

line 81 contamination is usually never by an anaerobe. This sentence is not useful and it would be better to discuss prior to this any negative controls that were used to ensure contamination was minimal.

line 87 why is there a) at the end?

line 106 "the very low level..." sentence is repetitive with the paragraph prior to this

line 172 you previously describe that the cells are using native carbon and not just the substrate added, as such this is expected that they communities don't match the substrate added. It may be good to reiterate this idea here.

line 517 add CDEBI publication number

Reviewer #3 (Remarks to the Author):

This manuscript reports on metabolic activity of microorganisms retrieved from sediments taken from an oligotrophic region of the ocean, the South Pacific Gyre (SPG). The authors present C and N incorporation data (using NanoSIMS), growth rates, and a DNA-based determination of the community identity. They conclude that microorganisms taken from microaerophilic sediments cores ranging in age from 13 to 101.5 million years old are not only capable of metabolic activity, but growth, and have been active on some level since deposition. This manuscript is another piece of evidence exploring the deep biosphere and therefore the limits to life. I support publication of this work. However, it could be clearer (comments below) and the authors should spend more text addressing two main points:

- 1) the anaerobic activity that they found, rather than dismiss it as a side note – in some ways it's more interesting since it's more of a microbiological feat.
- 2) how the incubations were such a departure from the sampling environment: the pressure of the environment; inclusion of a headspace; and years of storage since sampling.

A generic comment on the use of isotope labeling experiments:

For the labeled C incubations – if all of the carbon atoms in a compound are labeled (are they?), then for every mole of glucose taken up, 6 moles of labeled carbon are taken in. For bicarbonate its 1:1. Does this impact the scale of labeled uptake? A microbe would have to take up bicarbonate at 6 times the rate of glucose to show that same signal in the same amount of time, right? This might not explain all of the differences, but it could partly account for it and require a rescaling of the results.

Line 37 what is meant by “catch up in labeled nitrogen incorporation”?

Line 38 “anaerobic capabilities” should be changed to something like “microorganisms that can metabolize in anaerobic conditions”

Line 51 units are needed

Lines 65-66 “regions of interest” need to be defined – not everyone has a NanoSIMS

Line 68 ~3% O₂ in the gas phase in the headspace? Does this correspond to an aqueous concentration similar to in situ conditions?

Lines 74 –onward : what electron acceptors were available in the anaerobic conditions? For that matter, was the composition of the incubations monitored at all? Did CO₂ evolve? Were the sample anaerobic at the end of the incubation?

Line 77: what is meant by “hardly revived”? If they were revived at all, this is a big deal – after sitting in aerobic sediments for 65.5 Ma, they’re still capable of switching to anaerobic metabolism!? That’s incredible. It could be explained by an facultative aerobe, but it would still be something that the microbes could still switch after so much time.

Lines 96-98 this would be easier to read if it simply stated what is meant: Nitrogen incorporation was generally faster and more extensive than that for carbon.

Line 101: “which originally presented” means what? Do you mean, “which was originally present”?

Lines 111-112: should be “based on the rates of biomass and substrate...”

Lines 114-116 state that the microbes are using C and N from the porewater to grow, not the amended C and N. If this is the case, why aren’t the microorganisms this active in situ? There is no way that the growth rates measured ex situ could be sustained for very long in situ. This brings up another unstated assumption in this study – that laboratory measurements are relevant to the natural world. The incubations did not reproduce the pressure of the environment and they included a headspace – large departures from the natural system. Furthermore, the samples were taken in late 2010 and, presumably, stored for several years before these incubations were carried out. Even if a few contaminated microbes made their way into the samples at any point during sampling or storage, they could account for the bulk of the observed activity. These issues need to be discussed. I realize that there are limitations to do carrying out this type of research, but the manuscript would be strengthened by not only a full confession of them, but a discussion of how they could skew results.

Line 118 how can the organic carbon be bioavailable and protected? It’s either one or the other.

Lines 118-120 do you mean that the amount of organic carbon occurring in the samples is equivalent to the amount of carbon in 10¹⁰ cells per cm³? If so, state this. Also, how is this 10,000 times the amount needed to sustain growth, when a time period is not given? Organisms need energy per unit time.

Line 121 what is meant by “ecophysiological nature”? Do you mean how these microbes actually

behave in nature or the nature of their behavior or what? The phrase seems redundant.

Line 138 "have" not "has"

Lines 154-157 this is a cumbersome way of saying that despite earlier attempts (refs), the microbial diversity of SPG sediments 1 mbsf has not been determined

Lines 171-172 if the community composition is not related to how it responded to the substrate added, then what is the utility of the community analysis?

Lines 172-174 – this sentence restates the topic sentence of this paragraph eating up valuable space.

Figure 1 – give the age of the samples with their names; are the color bars numerical isotope ratios or ratios of measured intensities? Please state in caption.

Figure 2 put ages of samples in the gray bar in a);

a) the colors are too similar; try using a mixture of symbols and colors

b) the colors in incubation times 68 and 557 are too similar;

Caption – the first sentence is not helpful, delete it and raise the gray bar with the sample IDs above row a) so that it's clear that each panel under the name corresponds to data from that site; You don't need "Incubation durations indicated by color"

Figure 3 a closing ")" is missing;

a) the symbols are virtually unreadable; axes title should be shortened and units should be added to them

b) give ages of samples with their sample designation; again, the colors of many of phylogenetic groups are very similar; try patterns and colors

c) What is the NMDS plot telling us – that samples taken from the same site are similar? NMDS plots typically only impart obvious/already known information and do not reveal anything quantitative despite seeming to be so. They're fashionable, but what is the utility of a plot that would reveal the same 'information' with or without axes.

Reviewer #4 (Remarks to the Author):

The manuscript, "Aerobic microbial life that persists in oxic marine sediment for 101.5 million years", describes the results of a series of C and N incubation experiments performed on samples collected from the South Pacific Gyre (SPG). The authors follow their experiment for 557 days and measure incorporation of the labeled substrate with NanoSIMS. Overall, I found this to be an interesting study; however, I found the writing and presentation of the manuscript a bit difficult to follow.

One thing that I did not see in the manuscript were the age and storage conditions of the samples prior to incubation. Were the samples used in this study collected in 2010? What were the storage conditions of the samples prior to incubation? When were the incubation and NanoSims experiments performed? This information would be useful in the interpretation of the results.

Lines 67-70; Fig 2b,c. I initially had trouble interpreting the "bimodal" distribution and related discussion. I now believe that the authors are referring to the fact that given a labeled substrate, some cells may incorporate the labeled substrate while others may not. The authors try to illustrate this by using violin plots in Figure 2 and Extended Data 2. Overall, I think the violin plots are difficult to interpret whereas something like a beeswarm plot would be more interpretable (see

attached image). I feel that the beeswarm plots also highlight some of the more biologically interesting observations such as the rate at which cells respond to specific substrates and the proportion of cells that do not incorporate (or incorporate a very small percentage) substrate.

Lines 74-82. This paragraph was difficult to follow and should be rewritten for clarity. Additionally, with respect to the earlier comments about sample storage: could the authors comment on how conditions from collection to analysis may have changed? Could this have influenced the preservation of aerobic and/or anaerobic members of the community? Furthermore, is there a reason why a CH₄ incubation was not performed? One might expect to see CH₄ incorporation, especially in the anaerobic samples.

Lines 99-106. We are not informed as to what the C source is for the Ammonia samples in Figure 2b and the N source for the Acetate, Bicarbonate, Glucose, and Pyruvate samples in Figure 2c until this point. This makes the earlier parts of the submission difficult to follow/interpret. I think that submission would benefit by having the authors explain their experimental design earlier and improve their treatment labels and descriptions of Figures 1 and 2.

Lines 152-169. Is there a reason the taxonomic composition of the anaerobic incubations are not reported? Such an analysis may provide additional insight into the results presented for these samples.

Minor comments:

Line 45. As there have been more recent reviews of global biomass (Bar-On et al. 2018 and Magnabosco et al. 2018), one should be careful when reporting the relative contribution of the marine sediment biosphere to the global biomass. Bar-On et al. reports that marine sediment microbes account for ~12% microbial biomass and ~2% of the total living biomass. However, Bar-On et al. overestimates the contribution of the continental subsurface (Magnabosco et al.). Using the Magnabosco et al. continental subsurface estimate, I think that the contribution of marine sediments ends up being ~20-30% for microbial biomass, and ~1% for the total living biomass.

Line 51. What are the levels of O₂ found in the SPG? Cell concentrations are given but O₂ concentrations are not. This would also help interpret the author's choice of 3% O₂ for microaerobic conditions.

Lines 121-137. This section is difficult to follow and should be rewritten for clarity, especially regarding how these percentages are calculated.

Line 186. "absence" should be "abundance"

Figure 2. Given the fact that the authors often refer to ages associated with the samples, could the sample IDs be paired with the sediment age? (e.g. U1365 8H-2; 95.4 Ma)?

Figure 3b. As there are many categories with relatively similar categories, it is difficult to identify the proportion of some groups. Currently, the legend presents the taxa in alphabetical order but colors in the barplots do not follow this order. Having the order of taxa/colors in the plot and legend would make this figure easier to interpret.

Reviewer #1

Morono et al present a manuscript where they used cutting-edge stable isotope labeling techniques and single cell visualization of substrate utilization, from enrichment cultures deriving from deep seafloor oxygenated sediments that have ages ranging from 12 - 101.5 million years old. The conclusion is that the authors were able to revive bacteria from such ancient samples, which would indeed be truly stunning. However, in my opinion there is cause for concern regarding contamination, before these conclusions can be believed beyond a reasonable doubt.

We appreciate Reviewer 1's positive comments and useful suggestions, which improved the clarity of the manuscript. In the revised version of the manuscript, we have clarified the points of Reviewer 1's concerns as described below. (Please note that all the line numbers shown below are for the revised manuscript with marked-up changes.)

Unfortunately, there is no in situ data presented that is demonstrating presence of the enriched bacteria in the actual samples. This is needed to confirm that what grew was actually in the samples. Many of the bacteria enriched are related to typical contaminants. For example, Alphaproteobacteria, Betaproteobacteria and Firmicutes are possible contaminants (e.g., Bacillus spores from soil, Burkholderiales, and Rhizobium, they are ubiquitous in labs). Furthermore, there are a lot of cyanobacteria showing up in some of the samples - this must be contamination. How else you can explain that? In some cases the cyanobacteria are almost 50% of the sequences (e.g., 557 day timepoint). Some Chloroflexi are possibly authentic, since studies focusing on in situ communities of samples from the SPG and similar sites from the Atlantic showed that there should be a lot of Chloroflexi in these settings. Since the cell counts are very low for the authors sediments (10^2 - 10^3 cells gram) the possibility for contamination is extremely high. I know that the authors are using the best and most state of the art facilities and controls for contamination, but still the fact that what was enriched differs so much from the earlier SPG studies on in situ communities makes me wonder what happened over these long incubation times. Reading the methods, I have identified one possible source of contamination: the rubber stoppers and screw caps. It seems that they were not sterilized (at least this is not written). This alone could have introduced contaminants into all the incubations. Contamination from the outside of the rubber stopper can also be introduced when puncturing with a syringe (to add the labeled substrates). Its not clear how this was dealt with.

In their defense, it seems that the authors already looked at the OTUs they found in the enrichments and compared them against OTUs that they have sequenced from extraction blanks and or other sources of contamination in the lab. This is a good idea. That should also help to show how much of

what grew was contamination and how much was from the actual samples. I presume that most of the contaminants are Betaproteobacteria, Alphaproteobacteria, Firmicutes, and Cyanobacteria. If that is the case, how confident are you that OTUs from those same groups in your incubations are not contaminants? Are they at least from different genera? If they are all from the same genus, but different OTUs that is probably a contaminant in my opinion. If they are from different genera it is more convincing. There should be a long discussion on this in the manuscript.

Given these concerns, in my opinion there thus remains a critical aspect to be checked: do the substrate utilization rates observed by the authors derive from in situ microbes or those derived from bacteria introduced by contamination during experimental setup (e.g., from the non-sterilized rubber stoppers). The low rates of activity seen by the authors do not themselves serve as suitable check for this, because if you take a low number of contaminating bacteria and put them under nutrient depleted conditions then they should also grow slowly (or not at all).

Don't get me wrong, I think its entirely possible that some of the bacteria the authors observe could be from the actual samples (e.g., Chloroflexi) and actually be waking up after a long sleep. But, currently it is impossible to tell what is from the sample and what is a contaminant. Can the authors just do a 16S survey from DNA from the frozen samples to confirm that the OTUs that are enriched are also present in situ? That would basically provide the information needed, and satisfy all of the concerns I have outlined above.

Authors' reply

<Sterilization of materials used for incubation and contamination control of drilled core samples>

All of the incubation materials including rubber stoppers were sterilized by autoclaving before its use. Now complete description can be found in revised methods (lines 251-261 for experimental set-up and material sterilization, and lines 233-241 for contamination control of core samples).

<In situ community structure>

The microbial community structure shown as incubation time zero in Figure 3b of original manuscript shows the original microbial community structure of each sediment sample. Although we have tried DNA extractions and following 16S rRNA gene amplifications on frozen sediments multiple times, we had no success. Consequently, we used the strategy of separating microbial cells from sediment to avoid loss of DNA. For this separation, we used the samples of time zero, which were fixed and stored in PBS/EtOH solution. Although these were fixed samples and not perfect for DNA-based analysis, we could amplify 16S rRNA genes from

the sorted cells and identify the diverse microbes present in the original communities. We modified the description of this topic, which can be found in the revised text lines 169-180. Also, we now explicitly discuss the discovery of a cyanobacterial sequence in the revised manuscript (lines 209-217 in the revised manuscript)

<Microbial community structure of SPG sediments>

Please note that the microbial community composition of the oxic subseafloor sediments more than a meter below the seafloor is first reported in this study. Although previous studies showed the community composition in anoxic sediment outside the SPG¹⁹, near-seafloor SPG sediment (<10 cmbsf)²⁰, and SPG surface seawater²¹ (the reference numbers correspond those in the revised manuscript), diversity of microbial community in SPG sediment has not been reported. By separating microbial cells from sediment matrix, we could amplify and obtain the 16S-based community structures. Since the SPG sediment is oxic throughout the sedimentary column, it is natural that the community composition is different from those reported for anoxic subseafloor sediments.

Reviewer #2

This paper describes the incubations of deep oxic sediments and the observation that cells in these sediments readily take up substrates, suggesting that life is preserved in oxic sediments for long time periods. The experiments are well done and explained. This work adds to important understanding of the deep biosphere, but moreover raises more questions than it answers because few of the lineages recovered have any known long term resting form. Because of this, I think it is the proper level for a Nature Communications piece, since it should be widely read and cited for future work.

We thank Reviewer 2 for providing positive comments and suggestions, which significantly improved the manuscript. In the revised version of the manuscript, we have addressed all of Reviewer 2's comments and suggestions as described below. (Please note that all the line numbers shown below are for the revised manuscript with marked-up changes.)

Major comments:

The current discussion of controls is insufficient in the main text and should be improved.

Authors' reply:

We are not sure what type of control that the reviewer mentioned. We tried to modify the descriptions of experimental set-up, sterilization of material, and quality control of the samples in method section so that the readers can understand our control of quality in our experiments conducted. (lines 233-241 for quality control description and 395-412 for controls in DNA extraction and amplifications)

For the cells that did not respond to uptake, but appeared to remain dormant, what was their isotopic value? Do you have any idea on the ionic content? Recent work shows that LIBS is able to show that ions are lost when membranes are damaged - is there anything more that can be done to explain how/why these cells remain dormant?

Authors' reply:

Thank you for pointing out this important point. We think that the isotopic value of carbon and nitrogen for "dormant cells" has no detectable change along the time of incubation ("inactive ROIs"). They were observed in many of the incubation conditions but were apparent in the amino-acid-amended incubation of U1368D 2H-5. The cell concentration profile (first increase, and finally return to the initial cell concentration) that suggests that the fraction of microbes that grew upon addition of substrates probably died and were remineralized by small number of surviving cells at the later stage of the incubation. This gave the return of cell

abundance to close to the initial concentration and the disappearance of almost all the substrate-incorporating-cells. However, upon NanoSIMS analysis of the 557 day incubation sample, we still saw the “inactive ROIs” that was not remineralized like other cells and stayed unchanged throughout the incubation period, this is why we concluded that those are not the necromass, but the “dormant cells”.

The ionic content of microbial cells is certainly an interesting topic. However, the nature of sample preparation for NanoSIMS analysis requires fixation of cells and drying up completely to be placed in a high-vacuum analysis chamber (10^{-10} Torr). Upon initial fixation of cells, the membrane integrity and ions in the cells have already been lost. This means that we simply cannot observe the ionic content of the cells. It will be a highly interesting topic for future study to see the ionic content of potentially dormant cells in deep seafloor biosphere.

During uptake was any change seen in average cell size or shape?

Authors' reply:

Because our procedure to observe microbial cells (SYBR green I stain-based technique) has inherent risk of artificial inflation of size upon observation, we did not conduct in-depth determination of cell size for our incubated samples. As far as we could see during the microbial cell counts and NanoSIMS sample preparation, the size of the cells did not significantly change. Although we haven't systematically examined relationships of cell shapes to substrates added, almost all the cells in all experiments were spherules, with occasional short rods. Given this very limited morphological variation, we believe that cell shape analysis would provide little new information, in contrast to the phylogenetic information shown in Fig 3b.

I know there is a word limit, but noting the cyanobacteria observed in Ammonia incubations may be worthy.

Authors' reply:

According to the reviewers' suggestion, we revised the manuscript to include discussion about the discovery of cyanobacterial sequence (lines 199-207 in the revised manuscript).

I in general dislike the cell abundance graphs used. There are only a few datapoints and the trend between datapoints is truly unknown. If the authors can think of presenting them in any other way it would be useful.

Authors' reply:

We modified the presentation of the cell abundance graph. We understand the reviewer's concern, and now the data points are shown with dotted lines. We believe that the lines, on the other hand, are informative to understand the changes in cell abundance for each substrate along the time and are very important for our discussion. For the same reason, we changed presentation of the datapoints and partially transparent so that the individual points could be clearly recognizable. We hope this presentation will be acceptable.

Minor comments:

line 37 "catch up" is too informal

Authors' reply:

The description was revised

line 81 contamination is usually never by an anaerobe. This sentence is not useful and it would be better to discuss prior to this any negative controls that were used to ensure contamination was minimal.

Authors' reply:

According to the reviewer's suggestion, we deleted the corresponding sentence (lines 90-91 in the revised manuscript). We also modified the descriptions of experimental set-up, sterilization of material, and quality control of the samples in the method section so the readers can understand our control of quality in the experiments conducted. (lines 233-241 and 251-261 for quality control description).

line 87 why is there a) at the end?

Authors' reply:

Thank you for noticing this, we removed this unnecessary bracket.

line 106 "the very low level..." sentence is repetitive with the paragraph prior to this

Authors' reply:

We agree that the corresponding sentence is repetitive, although this was intended to discuss about ammonia metabolism. We removed the corresponding sentence.

line 172 you previously describe that the cells are using native carbon and not just the substrate added, as such this is expected that they communities don't match the substrate added. It may be good to reiterate this idea here

Authors' reply:

Thank you for this suggestion. We now include a sentence describing this. (lines 193-196

in the revised manuscript)

line 517 add CDEBI publication number

Authors' reply:

The number will be added after acceptance of this manuscript.

Reviewer #3 (Remarks to the Author):

This manuscript reports on metabolic activity of microorganisms retrieved from sediments taken from an oligotrophic region of the ocean, the South Pacific Gyre (SPG). The authors present C and N incorporation data (using NanoSIMS), growth rates, and a DNA-based determination of the community identity. They conclude that microorganisms taken from microaerophilic sediments cores ranging in age from 13 to 101.5 million years old are not only capable of metabolic activity, but growth, and have been active on some level since deposition. This manuscript is another piece of evidence exploring the deep biosphere and therefore the limits to life. I support publication of this work.

We appreciate Reviewer 3 for providing insightful comments and valuable suggestions, which significantly improved the manuscript. In the revised version of the manuscript, we have addressed all of Reviewer 3's comments and suggestions, and revised corresponding parts as described below. (Please note that all the line numbers shown below are for the revised manuscript with marked-up changes.)

However, it could be clearer (comments below) and the authors should spend more text addressing two main points:

- 1) the anaerobic activity that they found, rather than dismiss it as a side note – in some ways it's more interesting since it's more of a microbiological feat.*

Authors' reply:

We agree that anaerobic activity is surprising in this long-oxic sediment. Consequently, we have revised the manuscript to clarify this result. As we describe in the revised manuscript, although our incubation of the U1370 samples was set up without additional oxygen, the sediment contained ~1.5 μM of oxygen and introduced 16.4 nmol of oxygen (by porosity [73%] of the sediment) to the incubation vial, which is calculated to reach an equilibrated concentration of not more than 14.6 nM in the sediment. Although this concentration of 14.6 nM (0.46 ppb) is below the detection limit of the oxygen sensor (15 ppb in optode) and indistinguishable from anoxia, we must acknowledge that the starting condition was not anoxic. We added discussion of this issue in this revised manuscript (lines 75-90 in the revised manuscript)

- 2) how the incubations were such a departure from the sampling environment: the pressure of the environment; inclusion of a headspace; and years of storage since sampling.*

Authors' reply:

We started our incubations onboard during drilling expedition. As it can be seen in Methods section, we incubated subsampled mini-core in the incubation vial with addition of small volume (50~100 μ l) of substrate solution onto it to diffuse. The background idea for the incubation was not to disturb the natural structure of the sediment by slurring in media solution. The headspace was slightly oxic (~3.3% oxygen), to give similar concentration of oxygen as measured in the initial sediment. So we tried to minimize the departure of our incubation condition too much, although it is still not comparable to the in situ environment. However, we think that making slurry gives much more departure from the sampling environment than the incubation with mini-core.

A generic comment on the use of isotope labeling experiments:

For the labeled C incubations – if all of the carbon atoms in a compound are labeled (are they?), then for every mole of glucose taken up, 6 moles of labeled carbon are taken in. For bicarbonate its 1:1. Does this impact the scale of labeled uptake? A microbe would have to take up bicarbonate at 6 times the rate of glucose to show that same signal in the same amount of time, right? This might not explain all of the differences, but it could partly account for it and require a rescaling of the results.

Authors' reply:

As described in our methods section of our original manuscript, all of the carbon and nitrogen atoms are labeled (used isotopes were $^{13}\text{C}_6$ -glucose, $^{13}\text{C}_2$ -acetate, $^{13}\text{C}_3$ -pyruvate, ^{13}C -bicarbonate, ^{13}C - ^{15}N -amino acids mix [mixture of 20 Amino Acids], and ^{15}N -ammonium). Our incorporation rate calculation is based solely on the ratio to total carbon/nitrogen of the cells, which does not relate to any molar amount of substrate utilized. Also we use increase of biomass, as independent measure of growth of microbes. So, all the substrates are in same scale without the effect of the number of carbon nor nitrogen atoms in the molecule.

Line 37 what is meant by “catch up in labeled nitrogen incorporation”?

Authors' reply:

As per reviewers' suggestion, we modified the description of the corresponding part (lines 37-38 in the revised manuscript).

Line 38 “anaerobic capabilities” should be changed to something like “microorganisms that can metabolize in anaerobic conditions”

Line 51 units are needed

Authors' reply:

We made modifications to corresponding parts. (lines 38 and 53 in the revised manuscript)

Lines 65-66 "regions of interest" need to be defined – not everyone has a NanoSIMS

Authors' reply:

We put the description of region of interest in our method section. (line 335 in revised the revised manuscript)

Line 68 ~3% O₂ in the gas phase in the headspace? Does this correspond to an aqueous concentration similar to in situ conditions?

Authors' reply:

Our incubation vial initially contained ~3.3% of O₂ (we are sorry that the description in the original manuscript had inconsistent numbers of oxygen concentration, it was corrected in the revised manuscript), which should correspond to the aqueous concentration of oxygen at ~43 μM. The concentration of oxygen in the core samples (measured by oxygen electrode onboard during expedition) were ~70 μM for U1365C 8H-2 and 9H-3, ~150 μM for U1368D 1H-2, ~130 μM for 1368D 2H-5, and ~1.5 μM for U1370F 7H-6. We added detailed description of the oxygen concentrations of the core samples in method section of revised manuscript (lines 257-262 in the revised manuscript)

Lines 74 –onward : what electron acceptors were available in the anaerobic conditions? For that matter, was the composition of the incubations monitored at all? Did CO₂ evolve? Were the sample anaerobic at the end of the incubation?

Authors' reply:

The onboard analysis of pore water showed presence of nitrate (~46 μM) and sulfate (~27 μM) at the similar depth that the sample (U1370F 7H-6) was obtained. We noticed, on the other hand, that our incubation condition was set up without additional oxygen, but the sediment contained ~1.5 μM of oxygen and it introduced 16.4 nmol of oxygen (by porosity [73%] of the sediment) in the incubation vial, which is calculated to reach the equilibrated concentration of not more than 14.6 nM in the sediment. Even though the concentration of 14.6 nM (0.46 ppb) is below detection limit of the oxygen sensor (15 ppb in optode) and indistinguishable from anoxia, the starting condition was not anoxic, and we added discussion on this in revised manuscript (lines

75-90 in the revised manuscript).

Because the incubation set up used syringe-sampled mini-cores, not slurries, we did not have a way to verify that the incubation condition was anoxic with a redox indicator. Also, it was not possible to monitor the chemical conditions of the samples since 15 cm³ material was all used for fixation and the following microbiological analyses. And unfortunately, we did not monitor CO₂ in the headspace gas of the vial.

Line 77: what is meant by “hardly revived”? If they were revived at all, this is a big deal – after sitting in aerobic sediments for 65.5 Ma, they’re still capable of switching to anaerobic metabolism!? That’s incredible. It could be explained by a facultative aerobe, but it would still be something that the microbes could still switch after so much time.

Authors’ reply:

As in the reply to previous comment, we noticed that our incubation condition was that just did not add oxygen in the vial without adding reducing agent. So this is why that the low initial growth could be due to aerobes. However, the equilibrated oxygen concentration was not more than 14.6 nM and could be consumed quickly. Considering these issues, we modified corresponding discussion in the revised manuscript (lines 75-90).

Lines 96-98 this would be easier to read if it simply stated what is meant: Nitrogen incorporation was generally faster and more extensive than that for carbon.

Line 101: “which originally presented” means what? Do you mean, “which was originally present”?

Lines 111-112: should be “based on the rates of biomass and substrate...”

Authors’ reply:

Thank you for your useful suggestions. We modified the corresponding parts according to your suggestions. (lines 105-106, 111, 121-122 in the revised manuscript)

Lines 114-116 state that the microbes are using C and N from the porewater to grow, not the amended C and N. If this is the case, why aren’t the microorganisms this active in situ? There is no way that the growth rates measured ex situ could be sustained for very long in situ. This brings up another unstated assumption in this study – that laboratory measurements are relevant to the natural world. The incubations did not reproduce the pressure of the environment and they included a headspace – large departures from the natural system. Furthermore, the samples were taken in late

2010 and, presumably, stored for several years before these incubations were carried out. Even if a few contaminated microbes made their way into the samples at any point during sampling or storage, they could account for the bulk of the observed activity. These issues need to be discussed. I realize that there are limitations to do carrying out this type of research, but the manuscript would be strengthened by not only a full confession of them, but a discussion of how they could skew results.

Line 118 how can the organic carbon be bioavailable and protected? It's either one or the other.

Authors' reply:

We started our incubations onboard during the drilling expedition, so it was conducted by using freshly obtained core materials. In terms of incubation condition, we conducted our incubation by placing a mini-core in each incubation vial and by placing a small amount (~100 μ l) of substrate solution onto the mini-core to diffuse (detailed experimental set-up can be found in lines 247-274 in the revised manuscript). The idea of incubation with mini-cores is to keep the sedimentary environment for microbes as intact as possible, rather than slurring it with media, which should have given a larger departure from the natural system. However, although we paid careful attention to preserve the structure of the microbial environment and reproduce the *in situ* temperature, there were unavoidable differences in pressure. The water depths of the sites are respectively 5697, 3739, and 5075 meters below sea level (mbsl) at U1365, U1368 and U1370. Although we didn't measure the *in situ* pressures, they must slightly exceed the hydrostatic pressure at the seafloor, which ranges from roughly 37 ~ 57 MPa. Perhaps difference in pressure could have affected the bioavailability of organic carbon in our incubation but not *in situ*. According to Estes *et al.* (cited as reference #12 in the original manuscript), the organic carbon persists in oxic oligotrophic sediment through a combination of protective processes that involve adsorption to mineral surfaces and physical inaccessibility to the heterotrophic community. Perhaps the environmental pressure difference upon incubation opened the access of microbes in the sediments to the organic carbon being utilized. Modified discussion can be found in lines 121-133 in revised manuscript.

Lines 118-120 do you mean that the amount of organic carbon occurring in the samples is equivalent to the amount of carbon in 10^{10} cells per cm^3 ? If so, state this. Also, how is this 10,000 times the amount needed to sustain growth, when a time period is not given? Organisms need energy per unit time.

Authors' reply:

We just wanted to explain that the amount of bioavailable organic carbon is enough to sustain microbial growth from $10^2\sim 10^3$ cells cm^{-3} to 10^6 cells cm^{-3} , we modified the sentence to make things clear (lines 129-133 in revised manuscript)

Line 121 what is meant by “ecophysiological nature”? Do you mean how these microbes actually behave in nature or the nature of their behavior or what? The phrase seems redundant.

Authors’ reply:

We modified the description to “physiological status” to be clear.

Line 138 “have” not “has”

Authors’ reply:

Corrected

Lines 154-157 this is a cumbersome way of saying that despite earlier attempts (refs), the microbial diversity of SPG sediments 1 mbsf has not been determined

Authors’ reply:

We modified the corresponding part of the description to be more clearly understandable. (lines 169-180 in revised manuscript)

Lines 171-172 if the community composition is not related to how it responded to the substrate added, then what is the utility of the community analysis?

Authors’ reply:

Thank you for raising this point. We believe this result itself (that community compositions did not show clear relation to the substrate added) is very important. The utility of the community analysis is its necessity for determining this result (the lack of relation of community composition to the substrates added). And our results clearly suggest that the revival used not only the added substrates, but nutrients indigenous to the sediment samples.

Lines 172-174 – this sentence restates the topic sentence of this paragraph eating up valuable space.

Authors’ reply:

We rearranged the corresponding sentences. (lines 191-198 in the revised manuscript)

Figure 1 – give the age of the samples with their names; are the color bars numerical isotope ratios or ratios of measured intensities? Please state in caption.

Figure 2 put ages of samples in the gray bar in a);

a) the colors are too similar; try using a mixture of symbols and colors

b) the colors in incubation times 68 and 557 are too similar;

Caption – the first sentence is not helpful, delete it and raise the gray bar with the sample IDs above row a) so that it's clear that each panel under the name corresponds to data from that site;

You don't need "Incubation durations indicated by color"

Figure 3 a closing ")" is missing;

a) the symbols are virtually unreadable; axes title should be shortened and units should be added to them

b) give ages of samples with their sample designation; again, the colors of many of phylogenetic groups are very similar; try patterns and colors

Authors' reply:

We modified appearance of the graphs for clarity and visibility.

c) What is the NMDS plot telling us – that samples taken from the same site are similar? NMDS plots typically only impart obvious/already known information and do not reveal anything quantitative despite seeming to be so. They're fashionable, but what is the utility of a plot that would reveal the same 'information' with or without axes.

Authors' reply:

The NMDS plot showed that there was no clear dependency of microbial community composition on the substrates added for incubation. We understand that this has no quantitative information and we did not include any discussion in this regard.

Reviewer #4:

The manuscript, "Aerobic microbial life that persists in oxic marine sediment for 101.5 million years", describes the results of a series of C and N incubation experiments performed on samples collected from the South Pacific Gyre (SPG). The authors follow their experiment for 557 days and measure incorporation of the labeled substrate with NanoSIMS. Overall, I found this to be an interesting study; however, I found the writing and presentation of the manuscript a bit difficult to follow.

One thing that I did not see in the manuscript were the age and storage conditions of the samples prior to incubation. Were the samples used in this study collected in 2010? What were the storage conditions of the samples prior to incubation? When were the incubation and NanoSims experiments performed? This information would be useful in the interpretation of the results.

Authors' reply:

As described in the Methods, we started our incubations onboard during the drilling expedition, so it was conducted by using freshly obtained core materials. The incubation of the samples continued until around May 2012. Fixed samples were frozen at -80 °C and after storage, they were washed twice with PBS and preserved in PBS/ethanol (1:1 [v/v]) at -20°C until further processing. Further processing (separation of cells and sorting onto membranes) was performed until ~2014 in parallel to the NanoSIMS analyses. The NanoSIMS data processing and DNA extraction/sequencing continued for two more years, until we obtained the basic set of data in late 2016. In addition, it took several more years to obtain complete dataset that is included in this manuscript (including additional sorting of samples from stored sediment samples). Although this study took more time than we originally expected, we are sure that this long period of the work was necessary to conduct all the experiments in high quality since this is one of the most challenging low biomass environments on Earth.

Lines 67-70; Fig 2b,c. I initially had trouble interpreting the "bimodal" distribution and related discussion. I now believe that the authors are referring to the fact that given a labeled substrate, some cells may incorporate the labeled substrate while others may not. The authors try to illustrate this by using violin plots in Figure 2 and Extended Data 2. Overall, I think the violin plots are difficult to interpret whereas something like a beeswarm plot would be more interpretable (see attached image). I feel that the beeswarm plots also highlight some of the more biologically interesting observations such as the rate at which cells respond to specific substrates and the proportion of cells that do not incorporate (or incorporate a very small percentage) substrate.

Authors' reply:

Thank you for your suggestion. We changed our plot so that our results will be more interpretable.

Lines 74-82. This paragraph was difficult to follow and should be rewritten for clarity. Additionally, with respect to the earlier comments about sample storage: could the authors comment on how conditions from collection to analysis may have changed? Could this have influenced the preservation of aerobic and/or anaerobic members of the community? Furthermore, is there a reason why a CH₄ incubation was not performed? One might expect to see CH₄ incorporation, especially in the anaerobic samples.

Authors' reply:

We modified the corresponding paragraph for clarity. The sample storage time, as we explained earlier, was minimal and presumably had minimal influence on the microbial community. In terms of the anaerobic incubation, we noticed that the incubation condition for the samples obtained from U1370 7H-6 was the condition to be called as “nitrogen-flushed and without adding oxygen”. Although we didn't supply additional oxygen, the sediment initially contained ~1.5 μM of oxygen and it introduced 16.4 nmol of oxygen (by porosity [73%] of the sediment) to the incubation vial, which we calculate to reach an equilibrated concentration of not more than 14.6 nM in the sediment. Even though the concentration of 14.6 nM (0.46 ppb) is below detection limit of the oxygen sensor (15 ppb in optode) and indistinguishable from anoxic conditions, the starting condition was not anoxic. We added discussion of this point to the revised manuscript (lines 75-90 in the revised manuscript)

In terms of methane incubation, although we did prepare incubations with added methane, we excluded them from this study since there is no methane observed in the environment.

Lines 99-106. We are not informed as to what the C source is for the Ammonia samples in Figure 2b and the N source for the Acetate, Bicarbonate, Glucose, and Pyruvate samples in Figure 2c until this point. This makes the earlier parts of the submission difficult to follow/interpret. I think that submission would benefit by having the authors explain their experimental design earlier and improve their treatment labels and descriptions of Figures 1 and 2.

Authors' reply:

We added substrate information into the caption of Figure 2 (Figure 1 contained substrate

information), so readers can better follow and interpret the results.

Lines 152-169. Is there a reason the taxonomic composition of the anaerobic incubations are not reported? Such an analysis may provide additional insight into the results presented for these samples.

Authors' reply:

We added taxonomic composition of the incubation done without adding oxygen (U1370 7H-6) to Supplementary dataset 2.

Minor comments:

Line 45. As there have been more recent reviews of global biomass (Bar-On et al. 2018 and Magnabosco et al. 2018), one should be careful when reporting the relative contribution of the marine sediment biosphere to the global biomass. Bar-On et al. reports that marine sediment microbes account for ~12% microbial biomass and ~2% of the total living biomass. However, Bar-On et al. overestimates the contribution of the continental subsurface (Magnabosco et al.). Using the Magnabosco et al. continental subsurface estimate, I think that the contribution of marine sediments ends up being ~20-30% for microbial biomass, and ~1% for the total living biomass.

Authors' reply:

Thank you for this valuable suggestion. Now the number is revised. We included the results of Bar-On and Magnabosco (by including updated continental subsurface estimate) and revised the number to be "12~45% of total microbial biomass or 0.6~2% of total living biomass on Earth".

Line 51. What are the levels of O₂ found in the SPG? Cell concentrations are given but O₂ concentrations are not. This would also help interpret the author's choice of 3% O₂ for microaerobic conditions.

Authors' reply:

Our incubation vial initially contained ~3.3% of O₂ (we are sorry that the description in the original manuscript had inconsistent numbers of oxygen concentration, it was corrected in the revised manuscript), which should correspond to the aqueous concentration of oxygen at ~43 μM

measured shipboard. The concentrations of oxygen in the core samples (measured by oxygen optode onboard during expedition) were $\sim 70 \mu\text{M}$ for U1365C 8H-2 and 9H-3, $\sim 150 \mu\text{M}$ for U1368D 1H-2, $\sim 130 \mu\text{M}$ for 1368D 2H-5, and $\sim 1.5 \mu\text{M}$ for U1370F 7H-6. We added detailed description of the oxygen concentrations of the core samples in method section of revised manuscript (lines 257-262)

Lines 121-137. This section is difficult to follow and should be rewritten for clarity, especially regarding how these percentages are calculated.

Authors' reply:

The description was modified for clarity of general concept (lines 136-139 in the revised manuscript). Details of the calculation are unchanged from the original manuscript (lines 370-378 in the revised manuscript)

Line 186. "absence" should be "abundance"

Figure 2. Given the fact that the authors often refer to ages associated with the samples, could the sample IDs be paired with the sediment age? (e.g. U1365 8H-2; 95.4 Ma)?

Authors' reply:

Thank you for your suggestion. We modified our description accordingly.

Figure 3b. As there are many categories with relatively similar categories, it is difficult to identify the proportion of some groups. Currently, the legend presents the taxa in alphabetical order but colors in the barplots do not follow this order. Having the order of taxa/colors in the plot and legend would make this figure easier to interpret.

Authors' reply:

We noticed that the order of substrate shown in Figure 3 was not consistent with the other part (for example, Figure 3a) and modified the order. We changed some of the bar colors to be more easily visible to the reader. In addition, we double checked the order of colors, and confirmed that the order is now correct (in alphabetical order).

Reviewers' comments second round:

Reviewer #1 (Remarks to the Author):

In my original review of the manuscript, I requested a more detailed assessment in the text the influence of contamination but unfortunately this was not provided in the revised manuscript. Only one or two new sentences referencing the new table S2, which shows all the data including that from the extraction blanks and negative controls (I assume this is 'NA' in the table, no explanation provided by the authors).

I've downloaded the table and taken a look, and as I had suspected at least half the dataset includes OTUs that are present in the negative controls and are thus likely contaminant species. In the remaining half of OTUs that were not detected in the negative controls there are still plenty of typical contaminants there from human skin sources (*Acinetobacter*, *Streptococcus*). So, it seems that the negative controls were not sequenced with sufficient sequencing depths to detect all the contaminants. In order for the authentic (non contaminant) OTUs to be determined, a much more rigorous discussion and quality control process is needed.

Reviewer #3 (Remarks to the Author):

Since this is a re-review, I won't repeat my previous summary of the work. I don't have any major comments. I think that the following passages in the text require a bit of clarification before publication:

Abstract – line 35, should be “on average”

Figure 2 caption text, “For the carbon substrates are indicated by colors of bicarbonate, acetate, glucose, and pyruvate, ammonia was added as nitrogen source” doesn't make sense. Do you mean “Ammonia was used as the nitrogen source in the incubations that tested bicarbonate, acetate, glucose and pyruvate as the carbon sources”?

The paragraph on lines 191 – 198 says that the microbial community composition did not correspond to the identity of the substrate added. The text notes that “This is consistent with the suggestion from NanoSIMS-based results that the active microbial community utilizes native carbon and nitrogen compounds other than the substrates added to the incubation.” This is confusing because the main finding from this study is that the microbes are taking up labeled C and N compounds. Please clarify.

The following line in the captions for Extended Data Figure 2, should be modified to something like the following suggestion:

“For the incubated samples carbon substrates for incubation are indicated by color of bicarbonate, acetate, glucose, and pyruvate, ammonia was added as nitrogen source. The condition “Ammonia” was added only with ammonia, without addition of any carbon substrate.”

This should be something like:

“For the samples incubated with carbon substrates (bicarbonate, acetate, glucose, and pyruvate), ammonia was added as nitrogen source. The incubation labeled “Ammonia” received ammonia as the N source and no additional carbon.”

In the captions for figure 2 and Extended Data Figure 2, delete the first use of the word “The” in the last sentence.

Response to Reviewer 1

We thank Reviewer 1 for their recognition that our techniques are “cutting edge” and that the central claim of our manuscript, that living bacteria were revived bacteria from sediment aged 12-101.5 million years, is “truly stunning”.

We appreciate Reviewer 1’s concern for contamination in seafloor studies. However, we believe that their criticism of our revised manuscript in this regard is based on a mistaken reading of the data provided in the revised manuscript.

In my original review of the manuscript, I requested a more detailed assessment in the text the influence of contamination but unfortunately this was not provided in the revised manuscript. Only one or two new sentences referencing the new table S2, which shows all the data including that from the extraction blanks and negative controls (I assume this is 'NA' in the table, no explanation provided by the authors).

Authors’ reply:

We are confused by this comment. We revised the manuscript to add a detailed assessment of the quality of the samples for setting up the incubation experiments. Furthermore, the “time point 0” data in the manuscript (and in Table S2) constitutes the 16S survey data that Reviewer 1 said in their first review would meet their contamination concern. We discuss these points in greater detail below.

There are two critically different stages for potential "contamination" of a final dataset. The first stage (A) is during experimental set-up of the incubations. The second stage (B) is during DNA-based analysis (although there are several DNA extraction and sequencing steps that can be affected by contamination, they happen sequentially and we consider them as a single "stage").

In their original review, Reviewer 1 expressed concern for potential contamination at experimental set-up [stage (A)], as indicated in the following comment "*Given these concerns, in my opinion there thus remains a critical aspect to be checked: do the substrate utilization rates observed by the authors derive from in situ microbes or those derived from bacteria introduced by contamination during experimental setup*". As stated in our response to the first round of reviews, we revised our materials and methods section to include the details of contamination control at the stage of

experimental set-up of the incubations. In addition, it should be noted that we set up our incubation as mini-cores (as shown in Extended Data Figure 3), not as slurries. This incubation style is much less susceptible to contamination than slurry; even if a small number of contaminants were introduced into the incubation vial, contaminant growth would be restricted to the mini-core surface and would not dominate the community within the mini-core.

In their original review, Reviewer 1 said that a 16S survey of the pre-incubation sediment would meet their concern. "*Can the authors just do a 16S survey from DNA from the frozen samples to confirm that the OTUs that are enriched are also present in situ? That would basically provide the information needed, and satisfy all of the concerns I have outlined above*". As stated in our response to the original review, the requested 16S survey was included in the original manuscript as "time point 0" in Figure 3b. Similarly, the "time point 0" samples are shown as incubation time of 0 day, with the added substrate identified as NA (not applicable) in Table S2. The added substrate is identified as NA for the "time point 0" data in Table S2 because no incubation substrate was added to the sediment samples from which we extracted the DNA for the 16S survey of the in situ population.

I've downloaded the table and taken a look, and as I had suspected at least half the dataset includes OTUs that are present in the negative controls and are thus likely contaminant species. In the remaining half of OTUs that were not detected in the negative controls there are still plenty of typical contaminants there from human skin sources (Acinetobacter, Streptococcus). So, it seems that the negative controls were not sequenced with sufficient sequencing depths to detect all the contaminants. In order for the authentic (non contaminant) OTUs to be determined, a much more rigorous discussion and quality control process is needed.

Authors' reply:

It appears that Reviewer 1 has mistaken the reads from the real "time point 0" samples (pre-incubation sediment) for reads from "negative controls". According to their second review, Reviewer 1 assumed that NA identified this data as being from "extraction blanks and negative controls". We are sorry that our labeling of the data from these "time point 0" was unclear to the reviewer. As we mention above, NA refers to the absence of incubation substrate in the 16S survey of the pre-incubation sediment. It is reasonable that at least half of the dataset includes OTUs that are present in the reads

from pre-incubation sediment samples.

In revising our manuscript to address Reviewer 1's concern about contamination, we also added a detailed description of the "probabilistic" approach that we employed to remove potentially contaminating sequences from the sequence reads (lines 397-429 in the revised manuscript). Also, for the reviewer's information, we prepared Table X1, Figure X1, and X2 as review-only materials. As can be seen in Table X1, the majority of the sequences in controls (79% of extraction negative controls [ExNEGs] and 96% of PCR-no template controls [PCR-NTCs]) were removed by the probability-based contaminant-removal approach. The additional removal of sequences closely related to sequences found in "core microbiomes" from human body habitats (Li et al., PLoS One 8, e63139 [2013]) and typical contaminants (Glassing et al. [DOI 10.1186/s13099-016-0103-7]) had minimal effect on the number of the sequence reads in both samples and controls. As additional checks, we applied two additional conservative approaches: one in which all of the OTUs detected in controls were removed (most conservative removal [MCR] procedure) and one in which the OTUs detected more than 3 times in the controls (ExNEGs and PCR-NTCs) were removed (slightly relaxed removal [SLR] procedure). Although the MCR and SLR procedures removed large numbers of sequences from the samples, the basic ordination pattern did not change, and the arrows showing the tendency of the ordination stayed basically the same as with the probability-based contaminants removal. These additional checks demonstrate the robustness of our dataset and underscore the robustness of our results and conclusions.

Table X1 (review only) | Sequence reads and OTU numbers along the process of contamination removal. “All samples” indicates 73 samples (in total) shown in the main text and figures. Extraction negative controls, and PCR-no template controls were from 8 and 6 times trials, respectively. “Probability-based contaminant removal (PB)” was the method employed in this study to remove potential contaminants in the sequence results. The results of three additional sequence removal procedure tests were included. Bacterial sequences closely related to sequences found in “core microbiomes” from human body habitats (Li et al., PLoS One 8, e63139 [2013]) and typical known contaminants (Glassing et al. [DOI 10.1186/s13099-016-0103-7]) were additionally manually checked and removed from the sequence assemblage in PB+SGR procedure. In MCR, all of the OTUs that was detected in extraction negative controls (ExNEG) or PCR-no template controls (NTC) were removed. In SLR procedure, the OTUs that were detected more than 3 times in the controls (ExNEG and NTC) were removed.

	Before processing	Probability-based contaminants removal (PB)	PB-Removed sequence	PB + selected genera removal (PB+SGR)	PB+SGR-Removed sequence	Most conservative removal procedure (MCR)	MCR-Removed sequence	PB+Slightly relaxed removal procedure (SLR)	PB+SLR-Removed sequence
Number of sequences									
All samples	9960368	5918032	41%	5649046	43%	739768	93%	1132192	89%
Extraction negative controls	1061153	217794	79%	215106	80%	0	100%	9353	99%
PCR-no template control	803615	34690	96%	33431	96%	0	100%	3715	100%
Number of OTUs in samples	949	862		798		670		789	

Figure X1 (review only) | Phylogenetic composition of sequence reads in this study along the process of contamination removal. The phylogenetic composition at Phylum level (in Proteobacteria, it was shown at Class level) of “all samples”, “extraction negative controls (ExNEGs)” and “PCR-no template controls (NTCs)” after each of sequence removal procedure are shown. In “all samples”, in total 73

samples shown in main text and figures are included. ExNEGs are the blank extraction, which was conducted in parallel to the DNA extraction of the samples in which sterilized MilliQ water was used instead of the samples. PCR-NTCs were prepared by setting up PCR reaction (in parallel to the samples) by replacing samples with sterilized MilliQ water.

Figure X2 (review only) | Non-metric multidimensional scaling (NMDS) ordination plot based on Bray-Curtis dissimilarities of community compositions after four different sequence removal procedures. Only 73 “samples” shown in the Figure X1 was included in

these ordination plots. In the case of “MCR”, three samples (U1368 1H-2 ammonia 68 day, U1368 1H-2 bicarbonate 68 day, U1368 2H-5 ammonia 557 day) were excluded from the ordination plot.

Response to Reviewer 3's second review.

We thank Reviewer 3 for their comments in this second round of review. Addressing those comments improved the manuscript's clarity. We have revised the manuscript to follow their suggestions as described below. Please note that the line numbers shown below are for the revised manuscript with marked-up changes.

Since this is a re-review, I won't repeat my previous summary of the work. I don't have any major comments. I think that the following passages in the text require a bit of clarification before publication:

Abstract – line 35, should be “on average”

Authors' reply:

We corrected this line as Reviewer 3 suggests (line 34 in the revised manuscript)

Figure 2 caption text, “For the carbon substrates are indicated by colors of bicarbonate, acetate, glucose, and pyruvate, ammonia was added as nitrogen source” doesn't make sense. Do you mean “Ammonia was used as the nitrogen source in the incubations that tested bicarbonate, acetate, glucose and pyruvate as the carbon sources”?

Authors' reply:

In revising the manuscript, we modified this caption (lines 644-647 and the Extended Data Figure 2 caption [lines 678-681]) as Reviewer 3 suggests. Both captions now say “For the samples incubated with carbon substrates (bicarbonate, acetate, glucose, and pyruvate), ammonia was added as nitrogen source. The incubation labeled “Ammonia” received ammonia as the nitrogen source with no additional carbon.”

The paragraph on lines 191 – 198 says that the microbial community composition did not correspond to the identity of the substrate added. The text notes that “This is consistent with the suggestion from NanoSIMS-based results that the active microbial community utilizes native carbon and nitrogen compounds other than the substrates added to the incubation.” This is confusing because the main finding from this study is that the microbes are taking up labeled C and N compounds. Please clarify.

Authors' reply:

We thank Reviewer 3 for identifying this need for clarification. We have revised the text to clarify that the added substrates initially stimulated the revival of microbes and its activity for their utilization of native carbon and nitrogen compounds (lines 186-188 in the revised manuscript).

The following line in the captions for Extended Data Figure 2, should be modified to something like the following suggestion:

“For the incubated samples carbon substrates for incubation are indicated by color of bicarbonate, acetate, glucose, and pyruvate, ammonia was added as nitrogen source. The condition “Ammonia” was added only with ammonia, without addition of any carbon substrate.”

This should be something like:

“For the samples incubated with carbon substrates (bicarbonate, acetate, glucose, and pyruvate), ammonia was added as nitrogen source. The incubation labeled “Ammonia” received ammonia as the N source and no additional carbon.”

Authors' reply:

As we mentioned above regarding the Figure 2 caption, we identically modified both captions using the language that Reviewer 3 suggests.

In the captions for figure 2 and Extended Data Figure 2, delete the first use of the word “The” in the last sentence.

Authors' reply:

We have deleted this word in both captions as Reviewer 3 suggests.

REVIEWERS' COMMENTS third round:

Reviewer #1 (Remarks to the Author):

I thank the authors for clarifying the points in the manuscript, it seems my concerns regarding contamination were indeed related to a misunderstanding. In my original assessment, it was not clear to me that the T0 samples in Figure 3 were frozen in situ sediments, and it was not clear that the 'NA' samples in Table S2 were the in situ frozen sediments. It would be a good idea to explicitly state this in the figures and the figure legends that these are in situ frozen sample data presented. This satisfies my main concerns and I now recommend this manuscript for publication. I still wonder though, why no Chloroflexi and no Archaea in the in situ frozen samples? these are usually really abundant groups in the subsurface. But, perhaps after 100 million years it is just the Proteobacteria that can survive? Might be worthwhile to discuss. This is just a minor comment for the authors to consider prior to publication.

Reviewer #3 (Remarks to the Author):

I support publication of this manuscript.

Response to Reviewer 1's third review.

We thank Reviewer 1 for their understanding of our data. We have revised the manuscript to follow their suggestions as described below.

I thank the authors for clarifying the points in the manuscript, it seems my concerns regarding contamination were indeed related to a misunderstanding. In my original assessment, it was not clear to me that the T0 samples in Figure 3 were frozen in situ sediments, and it was not clear that the 'NA' samples in Table S2 were the in situ frozen sediments. It would be a good idea to explicitly state this in the figures and the figure legends that these are in situ frozen sample data presented. This satisfies my main concerns and I now recommend this manuscript for publication. I still wonder though, why no Chloroflexi and no Archaea in the in situ frozen samples? these are usually really abundant groups in the subsurface. But, perhaps after 100 million years it is just the Proteobacteria that can survive? Might be worthwhile to discuss. This is just a minor comment for the authors to consider prior to publication.

We added description about the T0 ("Time 0") samples in the legends for Figure 2 and Figure 3, to avoid any reader confusion. Also, we added explanation of the T0 samples and "NA" in the legend of Supplementary Data 2.

With respect to Chloroflexi, and Archaea – we detected very minor fractions of them (0~2.6% for Chloroflexi, and ~7% for Archaea) in our samples, including sediment before incubation. As we discussed in the previous rounds of review, no one had previously shown microbial community composition of South Pacific Gyre (SPG) sediment more than a meter below seafloor. Previously available knowledge about high abundances of Chloroflexi and Archaea was almost all from anoxic sediments and no previous information was available for general community composition in subseafloor SPG sediment. Although a recent paper by Vuillemin et al. (Science Advances 5, eaaw4108, 2019) showed abundant Chloroflexi and ammonia-oxidizing archaea in oxic subseafloor North Atlantic sediment (down to ~16 meters below seafloor), the geologic and oceanographic histories of these two regions are very different.

In terms of Chloroflexi, it might perhaps be reasonable to speculate that the obligate anaerobes were driven to extinction in SPG sediment by the very long interval of

exposure to oxygen (consistent with Reviewer 1's speculation that perhaps only Proteobacteria might survive 100 myrs of oxic burial).

A very recent paper on subseafloor Archaeal distributions reported Archaea to be relatively rare or absent in SPG sediment (Hoshino, T. & Inagaki, F. *ISME J.* 13, 227-231, doi:10.1038/s41396-018-0253-3, 2019). Furthermore, Vuillemin et al. reported decreasing Archaeal abundance with increasing sediment depth in the North Atlantic, which is consistent with them declining in abundance in oxic sediment on multi-myriadal timescales.

In summary, although the relative absence of Chloroflexi and low abundance of Archaea in our SPG sediment samples differs from the pattern observed by Vuillemin et al. (2019) for North Atlantic sediment, it is consistent with (i) the 101.5-Myr history of oxygen exposure in the SPG sediment, and (ii) recently published evidence that Archaea are generally rare in SPG sediment.